# Tropical Tropospheric Ozone Trends (1998 to 2023): New Perspectives from SHADOZ, IAGOS and OMI/MLS Observations *Accepted Version -20 Aug 2025*

**Anne M. Thompson[1,2]\*, Ryan M. Stauffer[1], Debra E. Kollonige[1,3], Jerald R. Ziemke[1,4], Bryan J. Johnson[5], Gary A. Morris[5], Patrick Cullis[5], María Cazorla[6], Jorge Andres Diaz[7], Ankie Piters[8], Igor Nedeljkovic[8], Truus Warsodikromo[9], Francisco Raimundo Silva[10], E. Thomas Northam[11], Patrick Benjamin[12], Thumeka Mkololo[13], Tshidi Machinini[13], Christian Félix[14], Gonzague Romanens[14], Syprose Nyadida[15], Jérôme Brioude[16], Stéphanie Evan[16], Jean-Marc Metzger[17], Ambun Dindang[18], Yuzaimi B Mahat[18], Mohan Kumar Sammathuria[18], Norazura Binti Zakaria[1], Ninong Komala[19], Shin-Ya Ogino[20], Nguyen Thi Quyen[21], Francis S. Mani[22], Miriama Vuiyasawa[22], David Nardini[23], Matthew Martinsen[24], Darryl T. Kuniyuki[5], Katrin Müller[25], Audrey Gaudel[24], Pawel Wolff[26], Bastien Sauvage[27]**

\*Corresponding author: Anne M. Thompson (anne.m.thompson@nasa.gov)

[1]NASA/Goddard Space Flight Center (GSFC), Greenbelt, MD, USA anne.m.thompson@nasa.gov; ORCID: 0000-0002-7829-0920; ryan.m.stauffer@nasa.gov; ORCID: 0000-0002-8583-7795
[2]University of Maryland-Baltimore County, Baltimore, MD 21228
[3]Science Systems and Applications, Inc., Lanham, MD, debra.e.kollonige@nasa.gov; ORCID: 0000-0002-6597-328X;
[4]Morgan State Univ., Baltimore, MD, jerald.r.ziemke@nasa.gov; ORCID: 0000-0002-5575-3654; [5]NOAA/Global Monitoring Lab, Boulder, CO; bryan.johnson2425@gmail.com; gary.morris@noaa.gov ORCID: 0000-0002-2196-8454; patrick.cullis@noaa.gov ORCID: 0000-0002-8723-6280; darryl.t.kuniyuki@noaa.gov; audrey.gaudel@noaa.gov ORCID: https://orcid.org/0000-0003-2727-213X
[6] Universidad San Francisco de Quito USFQ, Colegio de Ciencias e Ingenierías, Instituto de Investigaciones Atmosféricas, Quito, Ecuador; mcazorla@usfq.edu.ec; https://orcid.org/0000-0001-5295-2968
[7] Universidad de Costa Rica, San Jose, Costa Rica, jorge.andres.diaz@gmail.com
[8]Royal Netherlands Meteorological Institute (KNMI), De Bilt, The Netherlands, ankie.piters@knmi.nl, 0000-0002-8234-1022; igor.nedeljkovic@knmi.nl
[9]Meteorological Service of Surinam, Paramaribo, Surinam, wtruus98@gmail.com
[10] Brazilian National Institute of Space Research (INPE), Natal, Brazil, fraimundo.raimundo@inpe.br
[11] Science Systems and Applications, Inc., Wallops Island, VA., USA, e.thomas.northam@nasa.gov
[12] US Space Force Base, Patrick, FSB, FL, USA, Yang Enterprises, Inc., patrick.benjamin.1.ctr.gb@spaceforce.mil
[13] South African Weather Service (SAWS), Pretoria, South Africa, thumeka.mkololo@weathersa.co.za, tshidi.machinini@weathersa.co.za
[14] MeteoSwiss, Payerne, Switzerland, christian.felix@meteoswiss.ch, gonzague.romanens@meteoswiss.ch
[15] Kenya Meteorological Department, Nairobi, Kenya, syprosenyadida1@gmail.com
[16] Laboratoire de l'Atmosphère et des Cyclones (LACy), UMR 8105 CNRS, Université de La Réunion, Météo-France, Saint-Denis, France, jerome.brioude@univ-reunion.fr 0000-0002-5603-7924; stephanie.evan@univ-reunion.fr, 0000-0003-1014-0907
[17] Observatoire des Sciences de l'Univers de La Réunion (OSU-Réunion), UAR 3365, Université de La Réunion, CNRS, Météo France, IRD, Saint-Denis, France jean-marc.metzger@univ-reunion.fr
[18] Malaysian Meteorological Department, Petaling Jaya, Malaysia, ambun@met.gov.my, yuzaimi@met.gov.my, mohan@met.gov.my, norazura@met.gov.my
[19] National Research and Innovation Agency (BRIN), Bandung, Indonesia, ninongk@yahoo.com; ORCID: 0000-0003-3869-9622
[20] Japan Agency for Marine-Earth Science and Technology (JAMSTEC), Research Institute for Global Change, Yokosuka, Japan, ogino-sy@jamstec.go.jp ORCID: 0000-0001-9167-4833
[21] Aero-Meteorological Observatory (AMO), Viet Nam Meteorological and Hydrological Administration, quyenck@gmail.com
[22] The University of the South Pacific (USP), Suva, Fiji, francis.mani@usp.ac.fj, 0000-0001-6596-644x miriamavuiyasawa@gmail.com
[23] CIMAR, Univ. Hawaii at Manoa, Honolulu, HI; david.nardini@noaa.gov
[24] CIRES, Univ. Colorado, Boulder, CO, matthew.martinsen@noaa.gov

[25] Alfred - Wegener – Institut, Helmholtz Centre for Polar and Marine Research, Potsdam, Germany,
katrin.mueller@awi.de;  ORCID: 0000-0002-6891-6889
[26] SEDOO, Univ. Paul Sabatier III, Toulouse, France; now at ECMWF, Bonn, Germany,  pawel.wolff@ecmwf.int;
ORCID: 0000-0002-2082-6825
[27] Laboratoire d'Aérologie Observatoire Midi-Pyrénées, 14 av. E. Belin, 31400 Toulouse France;
Bastien.sauvage@univ-tlse3.fr;  ORCID :  0000-0003-3410-2139

Keywords: **Ozonesondes, Ozone Trends, Lower Stratosphere, Satellite Ozone, SHADOZ, IAGOS**

**Abstract.** Tropospheric ozone trends are important indicators of climate forcing and surface pollution
yet relevant satellite observations are too uncertain for assessments. The assessment project TOAR-II
has used multi-instrument, ground-based data for global trends over 2000-2022 (Van Malderen et al.,
2025a,b). For the tropics trends are derived from SHADOZ ozonesonde profiles (Thompson et al., 2021,
"T21"; Stauffer et al., 2024) or combinations of satellite, SHADOZ and IAGOS aircraft measurements
(Gaudel et al., 2024). We extend T21 that covered 1998-2019, analyzing SHADOZ data at 5 sites with a
Multiple Linear Regression (MLR) model for 1998-2023, and reporting trends for two free tropospheric
(FT) segments, lowermost stratosphere and total tropospheric column ($TrCO_{sonde}$). Trends for the Aura
period, 2005-2023, are computed from OMI/MLS $TrCO_{satellite}$. We find: (1) Extending SHADOZ analyses
four years shows little change from T21; $TrCO_{sonde}$ trends are small (0.5-1 DU/dec) except over SE Asia.
(2) Annual trends for $TrCO_{sonde}$ and OMI/MLS $TrCO_{satellite}$ agree within uncertainties with largest
differences at Samoa. Sensitivity tests show: (1) adding thousands of FT IAGOS profiles to SHADOZ
yields little change in trends; SHADOZ sampling is sufficient. (2) QR and MLR median trends are both
near-zero but QR captures extremes (5%-ile, 95-%ile) with changes up to $\pm$1 DU/decade (p < 0.10). (3)
Twelve-year analyses for trends lead to uncertainty changes too large for an assessment. This study and
Van Malderen et al. (2025a;b) provide the most reliable TOAR-II trends to date: over the past ~25 years,
tropical FT ozone changes are modest, ~(-3-+3)%/decade, except over SE Asia.

## 1 Introduction

The importance of tropical tropospheric ozone in atmospheric composition and climate variability
has long been known (Lacis et al., 1990; Schwartzkopf and Ramaswamy, 1993). Although the thickness
of total column ozone (TCO) in the tropics (~250-325 Dobson Units, DU; 1 DU=$2.69\times10^{16}cm^{-2}$) is much
less than in the extra-tropics (350-450 DU), the latitude band from $-30^{o}$ to $+30^{o}$ covers roughly 1/2 of
the Earth's surface. The tropical tropospheric column (TrCO) varies from ~15-20 DU over the Pacific to
50-60 DU over the Atlantic, giving rise to a distinctive wave-one pattern in both TCO and TrCO
(Thompson et al., 2003). Tropical tropospheric ozone is a major source of global OH (hydroxyl radical),
key to Earth's oxidizing capacity (Thompson et al., 1992), controlling the lifetimes of countless biogenic
and anthropogenic species. Global OH also controls the lifetime of methane, a powerful greenhouse gas
with both natural and anthropogenic sources (Khalil, 2000). Methane ($CH_4$) increases alone add ozone
to the troposphere and methane's oxidation by OH to carbon monoxide (CO), that also affects the
amount of OH, establishes a feedback cycle among $O_3$-OH-$CH_4$-CO (Thompson and Cicerone, 1986;
Thompson et al., 1990). Regional variability in factors controlling the cycle derives from local levels of
the shorter-lived nitrogen oxides and reactive volatile organic compounds. The tropics/subtropics ($-30^{o}$
to +30o) is where the "tropical pipe" (Plumb, 1996) carries ozone and ozone-destroying trace species
from the tropics into the mid-latitude lowermost stratosphere (LMS).
Trends in tropical tropospheric and LMS ozone are of interest for several reasons. First, free
tropospheric (FT) ozone is an important greenhouse gas. There is a potential for significant changes in
FT ozone because parts of the tropics are in areas of rapid changes in emissions. These may be caused
by economic development (Zhang et al., 2016) and/or variations in land-use and fire activity
(Christensen et al., 2022; Tsivlidou et al., 2023). Second, with relatively low ozone amounts relative to
the extra-tropics, ozone near the tropical tropopause is highly sensitive to dynamical interactions
Randel et al. (2007). Analyses of ozone profiles over the past ~25 years have found regional
meteorological changes propagating to seasonal ozone increases. Stauffer et al. (2024) verified that a
suspected decline in early-year convection (Thompson et al., 2021; hereafter T21) drove 1998-2022
ozone increases over equatorial southeast Asia. New reports on decreasing tropical cloud cover
(Tselioudis et al., 2025) and a shift in ITCZ location (Aumann et al., 2024) may indicate changes in
convection. At Réunion Island shifting anticyclones caused an increase in FT ozone from 1998-2023
(Millet et al., 2025). Recurring influences of climate oscillations, i.e., the Quasi-biennial Oscillation, ENSO,
Indian Ocean Dipole,  on FT and LMS ozone are well-documented in ozonesonde and satellite data
(Thompson et al., 2001; Ziemke et al., 2003; Ziemke et al., 2006; Ziemke et al., 2019; Lee et al., 2010;
Randel and Thompson, 2011; Thompson et al., 2011).

1.1        **The TOAR Project.  Challenges in Assessing Tropospheric Ozone Trends**

Context for this study comes from the International Global Atmospheric Chemistry/Tropospheric
Ozone Assessment Report (IGAC/TOAR) that is completing its second phase, TOAR II, initiated in 2020.
The first TOAR, designated here as TOAR I and published as a collection of 11 papers in *Elementa*, 2017-
2020; https://online.ucpress.edu/elementa/toar) included an assessment of surface ozone changes
(Chang et al., 2017) based on a vast set of surface ozone measurements from 7 continents (Schultz et al.,
2017). The TOAR I papers concluded that a comprehensive assessment of tropospheric ozone trends
based on observations is very challenging. For example, because the FT is the region of greatest
radiative forcing by ozone, the trends community needs profile data. A TOAR I evaluation by Tarasick et
al. (2019) pointed out the uneven geographic coverage of ozone profiles from soundings (~60 publicly
available station records since the early 1990s) and aircraft landing and takeoff profiles to ~250 hPa
that are used for FT ozone analyses. Tarasick et al. (2019) also questioned the suitability of all ground-
based (sonde, aircraft, lidar, passive spectrometers) for monitoring FT ozone using illustrations from
multi-decadal records that include obsolete techniques, evolving versions of instruments and/or
inconsistent absolute calibration. Nonetheless, several follow-on studies to TOAR I employed sonde
(T21) and commercial aircraft data (from the In-service Aircraft for a Global Observing System, IAGOS;
https://iagos.org), with 5-10% accuracy or better, to estimate trends from the 1990s to ~2018 (Gaudel
et al., 2020; Thouret et al., 2022).
Efforts in TOAR I to fill gaps with tropospheric ozone estimates from satellite data, preferred over in-
situ methods for their even global coverage, were mixed. Gaudel et al. (2018) pointed out that trends
derived from five satellite products covering tropics and mid-latitudes for the 2005-2016 period
differed from one another not only in magnitude but in sign. The newer TOAR II evaluation of six
satellite products for 2015-2019 over the tropics (Gaudel et al., 2024), where satellite estimates tend to
be most reliable (T21), exhibited a range of values. Not only were uncertainties among the products
highly variable; comparisons of monthly mean satellite columns with sonde and IAGOS profiles up to
270 hPa exhibited $r^2$ correlations as low as 0.27.
Other TOAR II studies also reveal a persistent uncertainty in the application of satellite data for
trends analyses. Pope et al. (2023) published global trends from 2005-2017 OMI (Ozone Monitoring
Instrument on board the NASA Aura satellite) TCO data that are too high because of a drift in OMI that
was corrected in Gaudel et al. (2024). The latter study led to a satellite-based estimate for tropical ozone
trends for 2005-2019 of ~(0.5-3) ppbv/decade for TrCO, similar or slightly larger than T21. Trends
determined from non-UV satellite instruments have been disappointing (Gaudel et al., 2018). A TOAR II
contribution by Froidevaux et al. (2025) examined changes in tropical ozone (±26 deg latitude) over the
2005 to 2020 period using measurements from the three lowest levels of MLS (Microwave Limb
Sounder on board Aura). Unfortunately, the zonal structure of MLS data at the 146 hPa and 213 hPa
levels (Fig. 1 in Froidevaux et al., 2025) shows that the prominent ozone wave-one feature, associated
with the Walker circulation (Thompson et al., 2003; Thompson et al., 2017), is absent, i.e., MLS does not
capture regional differences as OMI-based products do. Froidevaux et al. (2025; Figs. 1, 2, 5) compare
the MLS global structure to ozone output at 215 and 146 hPa from 3 models, Whole Atmosphere
Community Climate Model (WACCM6) and two variants of the Community Atmosphere Model with
Chemistry (CAM-Chem), each variant using different anthropogenic emissions. The models are likewise
zonally uniform in upper tropospheric ozone so they do not yield meaningful tropical upper
tropospheric ozone trends either.
A TOAR II-related investigation (Boynard et al., 2025) uses IR-based retrievals from the MetOP IASI
(Infrared Atmospheric Sounding Interferometer)  satellite instrument to determine trends for the
period 2008-2023. As in Gaudel et al. (2018) the newer IASI tropospheric ozone climatology differs from
that of UV-based products; IASI's negative ozone trends also disagree with increases from UV-based
products. Boynard et al. (2025) offer little explanation for the discrepancies. They note that their 12-yr
period trends (2008-2019), roughly half as long as the sonde or aircraft trends in T21 or Gaudel et al.
(2024), may be too short for a statistically robust result. Pennington et al. (2025) does provide some

insight into long-term changes of 3 satellite IR products (TROPESS CrIS, AIRS, AIRS+OMI) compared to ozonesondes and finds that the global tropospheric ozone satellite-sonde bias is approximately one third the magnitude of trends in global tropospheric ozone reported by TOAR I.

Keppens et al. (2025) addressed the question of whether satellite data harmonization for nadir ozone profile and column products improves satellite data consistency for both their mean distributions and long-term changes. They concluded that their harmonization methods reduce inter-product dispersion by about 10% when comparing to global ozonesonde datasets from 43 sites, but there is a significant meridional dependence, and the dispersion reduction is not consistent in space or time. This implies that a substantial part of the inter-product differences is instrument and/or retrieval-specific and the harmonization methods have limited application to TOAR II. An alternate method of combining the residual and profile products with a column fill-in method is expected to be published as the TOAR II "satellite assessment."

**1.2 TOAR II Studies with Ground-based Measurements. Statistical Issues.**

TOAR II has engaged a more globally representative set of researchers than TOAR I and reports data and analyses from a larger set of observations. Dozens of TOAR II-related publications can be reviewed at https://copernicus.org/articles/special_issue10_1256.html. Given the persistent uncertainty of the satellite records, a number of TOAR II contributors formed a community project, the Harmonization and Evaluation of Ground-based Instruments for Free-Tropospheric Ozone Measurements (HEGIFTOM), to apply newly standardized ozone measurement and processing protocols for data from sondes, aircraft and other ground-based (GB) instruments: FTIR, tropospheric ozone lidar and Umkehr retrievals from Dobson spectrometers. The rationale is that GB networks, with stable operations at fixed sites and well-calibrated instruments, e.g., as in the Network for Detection of Atmospheric Composition Change (NDACC; De Mazière et al., 2018), provide suitable time-series at dozens of sites over 7 continents and pole to pole. HEGIFTOM has two objectives: (1) harmonize data from ~80 long-term stations (1990s to 2023) in four GB networks using the most up-to-date reprocessing techniques with each record referenced to absolute standards; (2) calculate trends for the 2000 to 2022 period with harmonized data, reporting station trends with uncertainty. The trends for 55 individual stations are tabulated and illustrated in Van Malderen et al. (2025a; referred hereafter to as HEGIFTOM-1). Regional trends based on merging selected stations in densely sampled areas appear in Van Malderen et al. (2025b; referred hereafter to as HEGIFTOM-2).

Early in the TOAR II study period, in T21, we analyzed ozone profiles in the tropics collected in the Southern Hemisphere Additional Ozonesondes (SHADOZ) network (Thompson et al., 2003; Thompson et al., 2017) to compute FT and LMS ozone trends. The results in T21 are based on data from 8 combined SHADOZ stations within ±15 degrees latitude; the Goddard Multiple-Linear Regression (MLR)

model calculated trends from 1998 through 2019 from the surface to 20 km. Changes in individual
layers between 5 and 15 km were typically (5-10)%/decade, but only seasonally; over the equatorial SE
Asian stations at Kuala Lumpur and Watukosek, Indonesia, some layers displayed increases up to 20%/
decade (Fig. 6 in T21). However, in general, annually averaged changes in equatorial regions ranged
from ∼0 to +(1-2)%/decade or –(1-2)%/decade. In T21 LMS ozone (LMS defined as within 15-20 km)
computed with the MLR model displayed a seasonal loss up to 10%/decade (July through September)
or to 3%/decade, annually averaged. The loss maximized at ∼18 km. At the same time of year, a positive
trend in tropopause height (TH), derived from the SHADOZ radiosondes, was detected. Redetermining
LMS ozone changes in the 5 km column above the tropopause zeroed out the apparent trend.

More recently, the Stauffer et al. (2024, referred to as S24) paper demonstrated that over the 25-

yr period 1998-2022, early year (February through April/May) FT ozone increases in SHADOZ data
are associated with declining convection, most pronounced over SE Asia but observed to a lesser
degree at the other stations. With the newest OMI/MLS-based satellite estimates of total TrCO,
Gaudel et al. (2024) showed that over the Aura era (2005-2019) trends from satellite, SHADOZ and
IAGOS aircraft profiles were in good agreement with one another over southeast (SE) Asia, similar to
S24. SHADOZ only (T21) and IAGOS trends (Gaudel et al., 2024) diverge somewhat over the
equatorial Americas and Africa, partly due to a difference in sampling sites, e.g., west African IAGOS
profiles in Gaudel et al. (2024) versus the SHADOZ Nairobi station.
**1.3  This study**

We use the Goddard MLR model to calculate trends in monthly mean SHADOZ data for the

extended period 1998 through 2023, addressing the following questions:
-        Compared to T21, that reported on the 1998-2019 period, what do FT and LMS ozone
trends look like in the equatorial zone (-15° to +15°) with four additional years of SHADOZ
profiles? In other words, because the extension covers 2020 to 2023, the comparisons of 26-yr
trends with the 22-yr T21 record are looking for impacts of COVID-19 (Steinbrecht et al.,
2021; Ziemke et al., 2022).
-        How do trends in tropospheric ozone for 1998-2023 compare to trends for 2000 to 2023?
In other words, is the 26-year trend biased by SHADOZ starting at the end of the 1997-1998 ENSO
that produced strong perturbations to tropical ozone (Thompson et al., 2001)?
-        How do SHADOZ total tropospheric column (TrCO$_{sonde}$) changes compare to OMI/MLS
column ozone (Tr CO$_{satellite}$ ) trends over the 2005-2023 period? Do the satellite data capture the
seasonality of sonde-derived trends as noted in T21 and S24?

In addition to updated trends for equatorial SHADOZ sites, we have used SHADOZ profiles to

investigate three statistical issues raised in HEGIFTOM-1 and other TOAR II studies.
-    HEGIFTOM-1 calculated median trends for sondes and all other GB data using both Quantile
Regression (QR) and MLR models. Within the associated uncertainties of each, the two methods
gave identical results for annually averaged trends. Here we explore special features of QR and
MLR to learn more about the nature of the trends, e.g., seasonality (MLR), changes in the highest
and lowest quantile (QR).
-    Second, we use MLR to evaluate the sensitivity of trends on sample size as raised in TOAR II
papers (Chang et al., 2020; Chang et al., 2021; Gaudel et al. 2024) and HEGIFTOM-1 by
augmenting equatorial SHADOZ data with tropical IAGOS profiles for the appropriate region.
These calculations are carried out for FT ozone, i.e., in the region where radiative forcing is most
effective and both sondes and aircraft sample (700-300 hPa, roughly 5-10 km).
-    Satellite records are variable in length, with the most frequently used tropospheric estimates
starting after 2004 and a number of products merging measurements from multiple instruments.
We examine the sensitivity of trends to length of the observation period using QR by comparing
the 26-yr SHADOZ ozone trend to a recalculation that coincides with the IASI period 2008-2019
for which trends are reported in Boynard et al. (2025).
Data and analysis methods appear in **Section 2** with Results and Discussion in **Section 3. Section 4**
presents Summary and Conclusions.
**2.  Data and Methods of Analysis**
**2.1  Ozone datasets**
Three datasets are used in our study: ozonesonde profiles from the SHADOZ network; partial ozone
profiles from the IAGOS commercial aircraft network; monthly-averaged OMI/MLS tropospheric ozone
column estimates from 2005 through 2023.
**2.1.1  SHADOZ ozonesonde observations**
**Fig. 1a** displays the SHADOZ network stations, italicized with coordinates in **Table 1.** Quito (Cazorla,
2016; Cazorla et al., 2021; Cazorla and Herrera, 2022) and Palau (Müller et al., 2024) soundings, taken
since 2014 and 2016 respectively, have recently been added to the archive at
https://tropo.gsfc.nasa.gov/shadoz. The ozone profiles are obtained from ECC ozonesondes coupled to
standard radiosondes as described in earlier publications, e.g., Thompson et al., 2003; Thompson et al.
2007; Thompson et al., 2019. The profiles are archived with ozone uncertainties calculated with each
individual ozone partial pressure available as separate files at the SHADOZ archive (Witte et al., 2018;
WMO/GAW Rep. 268, 2021). Recent evaluations of ozonesonde data have established the quality of the
global ECC network. Measurements of TCO from 60 global ozonesonde stations average within ±2%
agreement with total ozone from 4 UV-type satellites since 2005 (Stauffer et al., 2022). About half of
SHADOZ stations exhibit a ~3-5% dropoff in stratospheric ozone (Stauffer et al., 2020) that is not
completely understood (Nakano and Morofuji, 2023; Smit et al., 2024). Accordingly, our study only uses
ozone data below ~50 hPa, defining the lowermost stratospheric (LMS) ozone in 15-20 km.
For the update to T21, that was based on 1998-2019 SHADOZ V06 ozonesonde data
(https://doi.org/10.57721/SHADOZ-V06), the same records for 8 equatorial sites, located between 5.8N
and 14S (color-coded in **Fig. 1b;** italicized in **Table 1**) are used with four additional years (2020-2023)
of ozone and P-T-U (pressure-temperature-humidity) profiles. These 8 stations have at least 14 years of
data between 1998 and 2023, although several have multi-year gaps (**Figs. 2, 3 Fig. S1**). For more
reliable statistics three of the "stations" or "sites" as they are referred to (**Fig. 1b**), are defined by
combining profiles from pairs of launch locations abbreviated as follows: SC-Para for San Cristóbal-
Paramaribo (dark blue dots in **Fig. 1b)**; Nat-Asc for Natal-Ascension (red dots in **Fig. 1b**); KL-Java (cyan
dots in **Fig. 1b**) for Kuala Lumpur-Watukosek (**Table 2**). T21 (see Supplementary Material) describes
multiple tests that were conducted to verify that these combinations are statistically justified. Annual
cycles in absolute column amounts (**Fig. 2**) and anomalies for the pairs were well-correlated. In T21
(Supplementary Material) total tropospheric columns integrated from sondes (TrCO$_{sonde}$) at the 8
individual stations were also well-correlated ($r^2$=0.72) with colocated TrCO$_{satellite}$ from OMI/MLS data
over the period 2005-2019. It is important to note that the 8 well-correlated sites are within 15 degrees
latitude of the equator. The correlation falls to $r^2$=0.50 when comparisons are made between sondes and
satellite columns for the 4 subtropical SHADOZ stations. FT ozone at those locations are seasonally-
dependent mixtures of tropical and extra-tropical air masses, with latitudes (**Table 4)** spanning Hanoi
(+21.0) to Irene (-25.9).
We have also analyzed trends of tropospheric ozone column and free tropospheric ozone at
individual SHADOZ stations using a QR model, following column definitions and guidelines for the TOAR
II/HEGIFTOM project analysis (Chang et al., 2023). The tropospheric ozone column in HEGIFTOM trends
analysis is defined as surface to 300 hPa; the FT is defined as a layer between 300 and 700 hPa and the
results are given as ppbv O$_3$/decade change and %/decade. The QR trends for 13 SHADOZ sites from
2000 to 2022 are summarized in **Table 4;** a subset of them appear in an evaluation of ground-based
global ozone trends in HEGIFTOM-1.
**2.1.2  SHADOZ and IAGOS-SHADOZ blended profiles. LMS and FT ozone**
The MLR trend analyses (results in **Table 2 and 3**) use SHADOZ profile measurements in several
ways.  First, the trends are computed using monthly-averaged ozone mixing ratios at 100-m intervals
from the surface to 20 km, as described in T21. Second, most results are illustrated as ozone column
amounts (in DU) for two FT segments, 5-10 km and 10-15 km, and for the LMS. Trends for ozone and P-
T-U data below 5 km are determined for completeness but are not tabulated because station sampling
times and local pollution can vary, giving artifact biases among the individual sites (Thompson et al.,
2014). We use 15-20 km for the LMS for two reasons. This is where several studies identified wave
activity associated with convection and ENSO-La Niña oscillations (Lee at al., 2010; Thompson et al.,
2011; Randel and Thompson, 2011; T21). Second, Randel et al. (2007) identified a distinct ozone annual
cycle in the 15-20 km range driven by the Brewer-Dobson circulation.
A third way of using SHADOZ profiles in the MLR analysis is in a blend with IAGOS aircraft profile
measurements within a lower FT pressure-defined region ("FTp" = 300-700 hPa, HEGIFTOM-1).
Calculations in the FTp segment are designed to add more samples within the SHADOZ-labeled
combination sites (compare profile numbers in **Tables 2 and 3**) and to augment regional trends in
HEGIFTOM-2 where no results are reported for the equatorial Americas, Atlantic Ocean or African
continent. In defining regions for merging SHADOZ and IAGOS observations, we follow locations
presented by Tsivlidou et al. (2023). Profiles from the SHADOZ Quito station (2014-2023) and two
IAGOS airports (Bogotá and Caracas) are added to the SHADOZ SC-Para profiles to define the equatorial
Americas for determining trends within the FTp (**Table 3**). Also, for the SHADOZ-IAGOS calculations,
sonde profiles from the Natal-Ascension pair are combined with 13 airports in west Africa (**Table 3**) to
determine trends for a region designated "Atlantic+West Africa," as shown in **Fig. 1b** (color-coded
circles) and the second column of **Table 3.** In "East Africa" Nairobi sonde data are combined with IAGOS
Nairobi and Addis Ababa ozone profiles. The FTp-designated Equatorial SE Asia consists of KL-Java
profiles from SHADOZ combined with IAGOS landing and takeoff data from Kuala Lumpur and
Singapore. Time-series of ozone column amounts (in DU and as anomalies) for SHADOZ stations and
airports for these 4 "regional" sites appear in **Fig. 3**. The coordinates of individual SHADOZ stations used
in the blended dataset (italicized) with IAGOS airports appear in **Table 1**. Calculations with FTp retain
Samoa as a single station.
**2.1.3 OMI/MLS satellite and sonde total ozone columns**
Trends computed with MLR for sonde-derived total tropospheric ozone columns (TrCO) are based on
integrating ozone mixing ratios from the surface to the thermal lapse-rate tropopause derived from the
radiosondes that accompany each ozonesonde launch. The standard WMO definition of tropopause is
used. For the 5 equatorial sites in our analyses, the tropopause is typically between 16 and 17 km. Our
$TrCO_{sonde}$ columns and trends are compared to $TrCO_{satellite}$, the tropospheric ozone columns estimated
from the OMI/MLS residual as described by Ziemke et al. (2019; updated in the TOAR II paper by Gaudel
et al., 2024). These newest OMI/MLR TrCO estimates have been corrected for a ~1%/decade upward
drift in OMI over the past two decades (Gaudel et al., 2024; SI material). The OMI/MLS column ozone
product is available starting in October 2004. We use monthly average TrCO for both sondes and
OMI/MLS between January 2005 and December 2023 **Fig. S1** in Supplemental Material. These are
identical to the data used in the Gaudel et al. (2024) TOAR II analyses of tropical ozone.
**2.2 Trend analyses**
**2.2.1 Multiple Linear Regression (MLR) model**
As in T21 and S24, the Goddard MLR model (original version Stolarski et al., 1991, updated in Ziemke
et al., 2019) is used for analysis of monthly mean ozone amounts. The MLR model includes terms for
annual and semi-annual cycles and oscillations prevalent in the tropics: QBO, MEI (Multivariate ENSO
Index, v2) and IOD DMI (Indian Ocean Dipole Moment Index; only for KL-Java):
$$O_3(t) = A(t) + B(t) + C(t)MEI(t) + D(t)QBO1(t) + E(t)QBO2(t) + F(t)IOD(t) + \varepsilon(t)$$
where t is month.  The coefficients are as follows: A through F include a constant and periodic
components with 12, 6, 4, and 3 month cycles, where A represents the mean monthly seasonal cycle and
B represents the month-dependent linear trend. When annual trends are reported, the B term includes
only the 12-month component to generate a single trend value over the period of computation. The
model includes data from the MEIv2 (https://www.esrl.noaa.gov/psd/enso/mei/), the two leading QBO
EOFs from Singapore monthly mean zonal radiosonde winds at 10, 15, 20, 30, 40, 50, and 70 hPa levels,
and IOD DMI (https://psl.noaa.gov/gcos_wgsp/Timeseries/Data/dmi.had.long.data).  The $\varepsilon(t)$ is the
residual, i.e., the difference between the best-fit model and the raw data. T21 noted that the monthly
ozone data and MLR model fits for the mid FT (5-10 km) and LMS layers are well-correlated. For the
LMS, for example, the correlation coefficients are r = 0.83-0.90 (**Fig. S7** in T21). The IOD DMI term is
included for KL-Java, the only station where the IOD impact on the ozone trend is reliably detected.
The 95% confidence intervals and p-values for each term in the MLR model as presented here are
determined using a moving-block bootstrap technique (10,000 resamples) in order to account for auto-
correlation in the ozone time series (Wilks, 1997). The model is applied to ozone anomalies in all cases
in order to minimize biases that might arise from intersite ozone differences between pairs for the
combined stations: SC-Para, Nat-Asc, KL-Java (**Table 2**). In other words, we calculate ozone anomalies
from the individual station's monthly climatology for all profiles before combining the pairs into
monthly means and computing the MLR ozone trends. Anomalies are also analyzed for the Nairobi and
Samoa station data, although this would be no different than computing MLR trends on the actual ozone
timeseries themselves. The MLR model was separately applied to the monthly mean ozone profile
anomalies at 100 m resolution, and the monthly mean partial column ozone anomaly amounts from 5-
10 km, 10-15 km, and 15-20 km. The MLR model was also applied to the monthly mean tropopause
height (TH) anomaly at each station, defined as the 380 K potential temperature surface (e.g., Wargan et
al., 2018).  Because TH and LMS ozone trends turn out to be strongly correlated (T21), the MLR analysis
was also performed for the ozone column amount anomalies referenced to the tropopause.  In that case
LMS ozone trends refer to changes in the 5 km above the tropopause with the FT extending from the
tropopause to 5 km below the tropopause (**Section 3, Table 2**). Finally, the MLR model was applied to
total tropospheric column amounts from the sondes (TrCO$_{sonde}$) and corresponding TrCO$_{satellite}$ from
OMI/MLS (surface to Tp in **Table 2)**.
Note that recent ozone trend studies and the TOAR II guidelines (Chang et al., 2020; Cooper et al.,
2020; Chang et al., 2023) have discouraged the use of nomenclature associated with statistical
significance whereas the Figures and Tables presented here refer to trends using terminology of 95%
confidence intervals (equivalent to p-value < 0.05), the most reliable results in **Section 3** (bold in
**Tables 2, 3** and **4)** are explicitly stated as based on p-values < 0.05.
Several studies of tropospheric ozone observations have noted a persistence of COVID-19
perturbations on ozone trends after 2019 (Ziemke et al., 2022; HEGIFTOM-1; HEGIFTOM-2). A
comparison of the extended SHADOZ mean ozone trends (1998-2023) relative to those from T21
(covering 1998 to 2019), both summarized in **Table 2,** represents the impact of COVID-19 in the deep
tropics. Likewise, SHADOZ was initiated at the end of the powerful 1997-1998 ENSO. Accordingly, we
applied MLR to the same 5 sites for 2000-2023 to evaluate any artifacts relative to the 1998 to 2023
trends. Those results also appear in **Table 2.**
**2.2  Quantile Regression (QR) model**
Whereas MLR has been the standard tool for analyzing global total and stratospheric ozone trends, the
latter often with satellite data where zonal means can be used, the TOAR II project has recommended
using QR as better suited for ozone trends in the troposphere where, for example, urban concentrations
can vary by factors of 3-4. Because it is a percentile-based method (Koenker, 2005), the heterogeneously
distributed changes of trends can be estimated, as shown, for example, in Gaudel et al. (2020).  To date
the TOAR II HEGIFTOM trends studies for observations at individual sites (HEGIFTOM-1) and regionally
organized data (HEGIFTOM-2) have been studied with the QR approach. In those studies and for the 13
individual SHADOZ time-series (**Table 4**) QR has been applied to the median change of the trends,
which is equivalent to the least absolute deviation estimator (i.e. aiming to minimize mean absolute
deviation for residuals; Chang et al., 2021). The rationale is that compared to least-squares criterion, a
median-based approach is more robust when extreme values or outliers are present. Median trends are
estimated based on the following multivariate linear model:
Observations[t] = a0 + a1*sin(Month*2π/12) + a2*cos(Month*2π/12)
+ a3*sin(Month*2π/6) + a4*cos(Month*2π/6) + b*t + c*ENSO[t] + N[t],          Eq. 1
where harmonic functions are used to represent the seasonality, a0 is the intercept, b is the trend value,
c is the regression coefficient for ENSO, and N[t] represents the residuals. Autocorrelation is accounted
for by using the moving block bootstrap algorithm, and the implementation details are provided in the
TOAR statistical guidelines (Chang et al., 2023).  In the individual site analyses of HEGIFTOM
observations (HEGIFTOM-1), where all individual ozone records (designated as L1) and monthly means
(denoted as L3) were analyzed, annually averaged trends turned out to be the same within
uncertainties. Where QR is applied in the present study, L1 ozone data are used.
**2.2.3  Trend Sensitivity Studies**
Three sensitivity studies were conducted, related to (1) sampling frequency; (2) the complementarity
of MLR and QR methods; (3) duration of time-series.
A number of TOAR-related studies (Chang et al., 2020; Chang et al., 2021; Chang et al., 2023) have
emphasized links between ozone time-series sampling characteristics, i.e., frequency of profile
measurements and/or temporal gaps, and trend uncertainty. Gaudel et al. (2024), for example, show
uncertainty (as 2-sigma) in median tropospheric ozone profiles; the inference is that 6-15 monthly
samples are required for meaningful FT trends (Fig. 1-2 in Supplemental Material, Gaudel et al., 2024).
For the first sensitivity test we examined trends dependence on sample size by comparing the annual
trends computed with MLR for the lower-mid-FT ozone segment (5-10 km) to the trends from the
combined SHADOZ-IAGOS merged monthly mean SHADOZ profiles (L3, 700-300 hPa) for 1998-2023. A
comparison of **Tables 2** and **3** indicate that for the equatorial Americas and Atlantic regions, the sample
numbers are increased more than a factor of 2; the other two sites have enhancements of 1.3 and 1.5.
The results in DU/decade and %/decade, appear in **Table 3.**
In the second sensitivity study three calculations were made using the data from all individual
SHADOZ stations as in HEGIFTOM-1, not only the 5-site equatorial profiles (**Table 2)**. The individual
station results appear in **Table 4.** The trend (1998-2023) in mean tropospheric column, TrCO (in
DU/decade), was computed using the QR method applied to all profiles at each station (L1, level 1 data)
with median (50%-ile) trends shown. The L1 sample numbers ranged from 326 (Watukosek) to 1142
(Hilo); the L1/L3 sample size ratios ranged from 2.2 to 4.3 with only 3 of 13 stations having a ratio
below 3.2 (i.e., fewer than 3 profiles per month). These trends can be compared to the MLR trends in
**Table 2.** The second calculation was a computation of trends at each station for the FTp column amount
used in HEGIFTOM-1(700-300 hPa, comparable to our 5-10 km segment in **Table 2** and the combined
IAGOS+SHADOZ FTp columns in **Table 3**). HEGIFTOM-1 pointed out the complementarity of applying
both MLR and QR to the 23-yr time-series of mean ozone column amounts. The advantage of MLR is a
graphical display of monthly trends that indicate important ways ozone interacts with seasonally
varying dynamics, as in S24 or Millet et al. (2025). QR distinguishes trends among different segments of
the distribution, an advantage for stations where tropospheric ozone segments are highly variable. To
demonstrate this, the QR method (including trends for the 5%-ile, 25%-ile, 75%-ile, 95%-iles) was
applied to IAGOS+SHADOZ combined FT dataset for the four regions: equatorial Americas;
Atlantic+West Africa; East Africa; equatorial southeast Asia.
The third sensitivity test investigates the degree to which trends and uncertainties depend on the
length of sampling. This issue arises because the most frequently used satellite estimates of
tropospheric ozone begin after 2003 compared to SHADOZ (1998-) and IAGOS (1994-). The Boynard et
al. (2025) study uses IASI products for 2008-2019 (and 2008-2023), comparing only 7 of ~40 potential
ozonesonde stations for evaluation. Using L1 mean TrCO tropospheric columns, the QR method was

applied to the 13 SHADOZ stations for a 12-year period (2008-2019) for comparison. The results in **Table 5** demonstrate the impact of sampling in a shortened time series.

## 3 Results and Discussion

### 3.1 Monthly and seasonal ozone climatology at 5 SHADOZ sites

**Figure 4** displays the 5-site monthly ozone climatology based on SHADOZ monthly averaged data from the surface to 20 km. Regional differences in vertical structure within the FT are pronounced. For example, the contours representing the 60-90 ppbv range (yellow to red colors) are absent in mid-FT ozone over KL-Java or Samoa (**Figs. 4d,e**). Conversely, FT ozone values ≤ 30 ppbv (darkest blue shades) observed over KL-Java and Samoa in the middle FT never appear over the other 3 stations: equatorial Americas (SC-Para, **Fig. 4a**), Nat-Asc or Nairobi (**Figs. 4b,c**). These contrasts may reflect regional differences in ascending vs. descending nodes of the Walker circulation. The latter feature is partly responsible for the tropospheric zonal wave-one (Thompson et al., 2003) that refers to a mean TrCO over the south tropical Atlantic Ocean that is sometimes twice as large as over the western Pacific. There is less regional variability in LMS ozone. At all stations (**Fig. 4**) above ~16 km the colors and contours are nearly uniform over the year. Mixing ratio contours of 100 ppbv and 200 ppbv may appear as a thick white line. The 100 ppbv level is sometimes referred to an ozonopause; typically it is within 1-2 km of the thermal lapse-rate tropopause.

### 3.2 FT and LMS ozone annual cycle (1998-2023)

The annual cycle of ozone at the two FT layers and for LMS ozone appear as anomalies in **Fig. 5.** FT ozone seasonality (**Figs. 5a,b**) is less uniform than for LMS ozone (**Fig. 5c**) and tropopause height (TH, **Fig. 5d**). Randel et al. (2007) showed that the near-uniform LMS ozone seasonality in the equatorial zone is due to the Brewer-Dobson circulation. The more varied FT ozone cycles in **Figs. 5a,b** are due to a range of different dynamical and chemical influences across the stations. As expected, the annual cycles for the pressure- and regionally defined FTp ozone (**Fig. 6** in %**)** resemble those for the corresponding SHADOZ sites in the lower (5-10 km) FT layer in **Fig. 5a**; the magnitudes are similar as well although **Figs. 5** and **6** are illustrated with different scales. In both cases it is seen that there are two seasonal maxima and minima for KL-Java (**Fig. 5a)** and equatorial SE Asia (**Fig. 6a**). The early year minima are associated with intense convective activity (T21, S24) that repeats in August at the onset of the Asian monsoon. KL and Watukosek are also affected by seasonal fire activity at the latter end of the rainy seasons. These features were described in detail in Stauffer et al. (2018) using Self-Organizing Map clusters and proxies for convection and fires.

### 3.3 FT ozone trends: regional and seasonal variability

#### 3.3.1 Trends for 1998-2023

In **Fig. 7** the trends in %/decade computed with MLR at 100-m intervals, for 1998 to 2023, are displayed (update of Fig. 6 in T21 for the 1998-2019 trends). Changes in the ozone column amounts for 1998-2023 computed from the model (DU/decade and %/decade) for the two FT layers (5-10 km, 10-15 km) appear in **Fig. 8**. A summary of values for the two layers (and for LMS ozone) appears in **Table 2.** The percentage values in **Fig. 7** and **Table 2** are the result of dividing the MLR B(t) term by the A(t) annual cycle of ozone term (Section 2.2.1). The MLR-calculated A(t) annual cycle derived from monthly mean ozone profiles (i.e., no anomaly calculation) is used to convert the B(t) trend in ppmv/decade (profiles) or DU/decade (partial columns) to %/decade. Ozone trends for both percent/decade and DU/decade are given in **Table 2.** Shades of red (blue) in **Fig. 7** represent ozone increases (decreases); cyan hatching denotes trends with p-values < 0.05. The annual mean trends in **Table 2** are computed by taking the average of the 12 monthly trends in DU and dividing by the mean seasonal ozone in DU to yield the annual percentage trend.

For 3 of 5 stations in **Figs. 8a** and **c**, there is a pattern of ozone increase at both FT layers in January to April. Percentage-wise the greatest increases are at KL-Java and Nairobi, ∼(10-15)%/decade in March and April. However, SC-Para and Samoa at 5-10 km (**Fig. 8a)** exhibit almost no trend at any time of year; at 10-15 km SC-Para and Nairobi show losses up to 10%/decade in February and (5-10)%/decade losses in August and September. However, **Table 2** displays no trend on an annual basis for SC-Para and Nairobi. Inspection of **Fig. 7** suggests small FT trends at Nat-Asc; **Table 2** displays a +3.4%/decade increase in the 10-15 km layer from 1998-2023. The total column, integrated to the tropopause, TrCO (**Fig. 9**) over Nat-Asc, has increased (1.9± 1.8)%/decade, p<0.05. There are no other annually averaged trends in the FT layers but TrCO for KL-Java (KL-Watukosek in **Table 2**) also increased, (2.6± 2.3)%/decade.

### 3.3.2  FT ozone trends sensitivity to COVID-19 and 1997-1998 ENSO

A comparison of the **Table 2** columns for 1998-2023 relative to those for 1998-2019 (the latter is from T21) reveals little. Only the 10-15 km layer at Nat-Asc has entries with p<0.05 for both periods. The extra 4 years reduced the positive trend slightly.  This is consistent with studies that found lingering COVID-related declines in sondes and satellites (Ziemke et al., 2022; HEGIFTOM-1). In **Table 2** columns for trends for 2000-2023 can be compared to those for 1998-2023. There is little information in the 2000-2023 column, i.e., no trends anywhere except for the TrCO for KL-Java, an area that was well-studied with satellite and some sonde measurements for the period affected by the large ENSO, amplified by the Indian Ocean Dipole pattern (Thompson et al., 2001). After August 1997, as a result of exceptionally high fire activity, ozone increased greatly. That could have meant a smaller change between ozone levels from 1998 through 2023 which would be consistent with a larger, more robust trend for 2000-2023 (4.6%/decade for KL-Java) compared to T21, 2.6%/decade (both p<0.05). Similar

trend differences for Kuala Lumpur are also observed with the QR trends in HEGIFTOM-1 (~4-
5%/decade for 2000-2022) versus 2.7%/decade in **Table 4**.

**3.4  LMS ozone trends and mean vertical trend over 5 SHADOZ sites**
In T21 (Figs. 10 and 11) trends in the LMS (nominally 15-20 km) showed 5-10%/decade decreases for
Nat-Asc, KL-Java and SC-Para between July and October. For the same months those locations exhibited
a tropopause increase ~100 m/decade, suggesting that the seasonal ozone increase is an artifact of a
changing tropopause. In other words, if the TH increased more air with relatively lower ozone would be
located in the 15-20 km layer. We tested this hypothesis by recomputing ozone column changes
referenced to the TH for 1998-2019, i.e., evaluating trends in a 5-km thick layer above the TH. The result
was that the apparent loss of LMS ozone from July to September or October disappeared. The same
analyses performed with LMS ozone and TH for the 1998-2023 period (**Fig. 10)** are the same as for
1998-2019 (T21).
Whatever the cause(s) of ozone loss in the LMS, it is a feature clearly captured by SHADOZ data as seen
in annually averaged ozone trends derived from the analyses displayed in **Fig. 11.** At 18 km the
composite trend from the 8 SHADOZ stations analyzed with MLR is (-4±3)%/decade. The mean trend
from ~13 to 3 km is zero, albeit with a ±2σ (95%) ±%/decade. Only below ~2km is the mean ozone
trend clearly positive. Most of that increase originates from near-surface pollution over equatorial SE
Asia (Fig. 6 in S24).
**3.5  Sensitivity tests.  Trends method.  FT sample numbers.  Length of time-series**
**3.5.1  Complementarity of Trends Methods**
In **Table 4** the median (50%-ile) QR trends from 1998-2023 for the TrCO and FTp ozone segments
for the 13 individual SHADOZ stations are presented.  The trends for the tropical stations are
comparable to the MLR trends in **Table 2 and 3**, respectively, within their uncertainties, reaffirming the
important HEGIFTOM-1 conclusion, i.e., MLR and QR trends from ground-based data (FTIR and Umkehr,
as well as sondes) are essentially the same. In **Fig. 12** the timeseries and histograms show the
distribution of IAGOS+SHADOZ ozone anomalies (in DU) for the four regions. The median trends from
1998 to 2023 (50%-ile) for the FTp ozone segments are also displayed with the lowest and highest (5%,
95%, respectively), 25%-ile and 75%-ile quantiles with red circles denoting p< 0.10. The medians are
statistically nearly the same, although as expected, the 2-σ uncertainty bars are smaller with the QR
method than with MLR. The MLR trends are higher in all cases except the equatorial Americas (**Fig.
12a**). In the latter case, the positive anomalies have increased significantly for the 5% and 25%
quantiles over the 26-yr period with no change for the 50%, 75% or 95% quantiles. This signifies that
the background (lowest-ozone) air has increasing ozone but the highest-ozone distribution has not
changed. Over the Atlantic+West Africa (**Fig. 12b)** there are also small increases in lowest part of the

distribution but the median and higher %iles show no change. Note Natal alone (when not combined
with other stations) shows high confidence increases in FTp (and TrCO) ozone (**Table 4**). East Africa
(**Fig. 12c**) shows marked increases in the lowest-ozone quantiles and the median but a significant
decrease in the highest-ozone (more polluted) air. The opposite is true over equatorial southeast Asia
(**Fig. 12d**). The most polluted air corresponds to ozone increasing but the background (lowest ozone)
FT segment shows an ozone decrease.

### 3.5.2   FT sample numbers

**Figure 13** monthly ozone trends are based on a total of 1.8 times the number of profiles as those in
the other lower FT layer, **Fig. 8a** and **8b.** For the equatorial Americas (blue) twice as many profiles
contribute to the trends in **Fig. 13** than in **Figs. 8a** and **8b;** for the Atlantic+West Africa includes 2.5
times more profiles than in **Figs. 8a** and **8b.** In all four regions, the seasonality of the trends is nearly the
same between the 5-10 km FT segment (**Figs. 8a** and **8b)** and the corresponding SHADOZ+ IAGOS trend
**(Fig. 13).** Furthermore, month by month, the trends are similar to those in the 5-10 km layer in both
magnitude and confidence level (uncertainty). **Table 3** shows no trends ($p < 0.05$) at any location. The
null trends are illustrated in the annual means at the right of each image in **Figs. 8** and **13**. This was
unexpected given the Chang et al. (2023) and Gaudel et al. (2024) suggestions that the uncertainty
should decline with more samples and positive trends might be amplified.

### 3.5.3    Length of time-series

**Table 5** displays a comparison of TrCO trends determined by QR for the SHADOZ 26-yr period, 1998-
2023 for 13 individual SHADOZ stations, and for the same time-series only between 2008 and 2019. The
uncertainties (expressed as ±2-σ) increased by factors of 2-3 or more for 10 of 13 stations. These results
are based on a Comment on Boynard et al. (2025) that also compared the time-series changes for 17
mid-latitude ozonesonde stations. Similar uncertainty increases were noted for all 27 stations. Of that
total, 9 stations exhibited ozone trend sign changes with the 12-year time-series although only 10
trends were statistically significant. In **Table 5** 2 stations change sign in their trends, 3 stations have a
change in confidence level ($p < 0.05$); note that 8 of 13 stations exhibit a substantial change in their trend
value, e.g., Samoa, Ascension Island. These results reinforce the need for multi-decade time-series.

### 3.6 Total tropospheric ozone trends, TrCO (1998-2023), from OMI/MLS and SHADOZ

Trends for the most recent version of OMI/MLS $TrCO_{satellite}$ are based on monthly mean satellite data
and determined with MLR over the period 2005 through 2023. Trends for total tropospheric column
ozone ($TrCO_{sonde}$) at the 5 SHADOZ sites for the same period appear in circles on the map in **Fig. 14**
where the stippling indicates no trend can be determined. For both OMI/MLS and the sondes (**Fig. 7**)
shades of red indicate total column ozone increases; blue represents declining ozone over the period of
analysis.  The mean annual $TrCO_{sonde}$ trends appear in the two rightmost columns in **Tables 2** and **6.** In

**Fig. 14** OMI/MLS shows trends > 1 DU/decade (typically 2-9%/decade) only appear over equatorial SE
Asia and parts of South America and the eastern Pacific at ~5N latitude. Circles indicate locations and
trends for the individual SHADOZ stations. The SHADOZ trends display lower trends than OMI/MLS. On
a month-by-month basis, the sonde and OMI/MLS trends are compared in **Fig. 15.** In 3 cases the
seasonality of TrCO trends from sonde and OMI/MLS are similar and the annually averaged OMI/MLS
TrCO$_{satellite}$ trends are not different from zero (symbols at right of each image). The seasonality of the KL-
Java monthly trends agrees well with OMI/MLS; the satellite mean is +5%/decade, gray in **Fig. 15d**. The
sonde SC-Para trend (**Fig. 15a; Table 6**) is quite a bit lower early in the year than the OMI/MLS trends
over San Cristóbal and Paramaribo that average +(2-3)%/decade. The Samoa sonde trend and OMI/MLS
TrCO trends diverge most of the year. The satellite annual trend is close to +10%/decade, an outlier
globally (HEGIFTOM-1) as well as over the tropics where no ground-based study results in a significant
trend for TrCO$_{sonde}$ over Samoa (**Section 4.2**).
**4  Summary and Conclusions**
We have presented a two-part evaluation of tropical ozone trends using a 26-yr record of
ozonesonde (SHADOZ, 1998-2023) profiles with selected FT aircraft (IAGOS) ozone data and the most
recent OMI/MLS estimates of tropospheric column ozone for 2005-2023. The next section summarizes
the findings. It is followed by **Section 4.2** which compares our trends to related TOAR II studies. **Section**
**4.3** concludes with a consensus view of FT ozone trends and perspectives relevant to the overall TOAR-
II climate and tropical assessments.
**4.1  Summary of findings**
The first part of the study updates trends in FT and LMS ozone for 5 stations, Nairobi, Samoa and
three combination sites (San Cristóbal-Paramaribo, Natal-Ascension, Kuala Lumpur-Watukosek)
extending the T21 trends, that covered 1998-2019, by 4 years. The new analysis added monthly
averaged data from 2000 to 2023 to the Goddard MLR model with standard proxies. Trends in FT (5-
10km, 10-15 km) and LMS (15-20 km) layers are illustrated with monthly means and annually averaged
changes in DU/decade and %/decade. Trends determined for the period 2000-2023 assessed impacts of
the 1997-1998 ENSO on a possible anomalous starting point. Comparisons of trends for the monthly
averaged Aura-derived OMI/MLS total tropospheric column ozone product (TrCO$_{satellite}$) were made to
those from monthly sonde-derived TrCO$_{sonde}$ for 8 equatorial SHADOZ stations. The principal results of
the SHADOZ trend updates and comparisons are as follows:
• The overall characteristics of T21 trends in the FT and LMS are confirmed with 4 additional years
of SHADOZ observations. From 1998 to 2023, regional and seasonal variability remains
pronounced with FT ozone increasing in thin layers at 4 of 5 SHADOZ stations ~(5-20)%/decade,
mostly between January and May. The exception is at SC-Para where there was a 5-10% ozone
decrease between 10-15 km during 1998 to 2023 compared 5-10%/decade increases in 1998-

2019 (T21). For 1998-2023, the greatest ozone increases occur in multiple layers below 10 km

over Nairobi and KL-Java and between 10-15 km over Samoa. However, these features do not

translate into annually averaged trends ($p<0.05$) in the 5-10 km or 10-15 km segments except

over Nat-Asc, i.e., adding 4 years of data to equatorial SHADOZ data does not modify the T21

picture of little or no FT ozone change. Only when the total tropospheric column ($TrCO_{sonde}$) trend

is evaluated do Nat-Asc (1.9±1.8)%/decade and KL-Java (2.6±2.3)%/decade) exhibit the slightest

trend ($p<0.05$). Examining the 5-station average in vertical form shows a null trend from ~3 to

17 km (0±2% within 2$\sigma$ up to 7 km and ~0±3% from 7 to 17 km). The marginal overall mean

increase, +5%/decade below 3 km, is primarily driven by KL-Java changes.

• With the starting year delayed to 2000, the $TrCO_{sonde}$ KL-Java trend (2000-2023) is almost twice

as large as for 1998-2023, indicating an effect of the 1997-1998 ENSO on equatorial SE Asia. This

is not surprising. Watukosek soundings (1997-1998) show ENSO-induced anomalously high

ozone over Indonesia that was also captured by satellite tropospheric ozone estimates from

TOMS (Thompson et al., 2001).

• The T21 LMS ozone and TH trends are also confirmed with 4 more years of data. For the layer 15-

20 km, ozone losses ~5%/decade from June through October, on average, give an all-site average

of -3%/decade at 17.5 km, a value similar to satellite averages (Godin-Beekmann et al., 2022). As

in T21, re-determining the LMS trends for an ozone column 5 km above the tropopause from

1998 to 2023, causes the trend to disappear.

• Annually-averaged trends, 2005-2023, determined with MLR for OMI/MLS columns, $TrCO_{satellite}$,

over the 8 individual equatorial SHADOZ stations (members of the 5 combined sites) and

$TrCO_{sonde}$ overlap within the uncertainties of each. Trends are close to zero at Nat-Asc, Nairobi

and KL-Java. The OMI/MLS $TrCO_{satellite}$ trends are marginally positive at SC-Para, with monthly

cycles diverging in the early part of the year at SC-Para. OMI/MLS trends do not capture large

monthly seasonal variations seen in SHADOZ profiles, especially for negative trends. The large

positive trend from OMI/MLS over Samoa does not align with determinations of FT ozone from

this or other studies (see below).

The second part of the investigation was motivated by statistical issues raised in related TOAR II

trends analyses. The results of these analyses are summarized:

• Trends methods. The relative merits of trends computed with QR and MLR, previously

demonstrated in HEGIFTOM-1, were reinforced with analysis of combined FT SHADOZ-IAGOS

data. Although median trends are the same, QR uncertainties are smaller. MLR is superior for

capturing seasonal influences but QR provides vital information on whether the background, low-

ozone or high-ozone (polluted) populations are changing the most.

- Sampling frequency. The sensitivity of the 1998 to 2023 FT ozone trends to sample number was explored by using IAGOS profiles to increase the SHADOZ sample size for the equatorial stations by 80% overall, including a doubling over the equatorial Americas and Atlantic regions, then applying MLR. Median trends were nearly unchanged. No FT trends over the 4 regions (plus Samoa) are detected with $p < 0.05$ although uncertainties, expressed in ppbv/decade, improved 30% over the equatorial Americas and by ~15% over 3 of the 4 other sites. These results indicate that current SHADOZ sampling is sufficient in this radiatively-important region.

- Length of trends. These were examined for the individual station TrCO$_{sonde}$ trends by comparing trends for 1998 to 2023 with a 12-year trend (2008-2019), one of two scenarios investigated by Boynard et al. (2025). The uncertainties (as 2-σ limits) increase by a factor of 2-3 for TrCO at 10 of 13 SHADOZ stations and some median trends change sign compared to the 26-year trends (**Table 5**). The next section shows that even 16-year IASI/Metop trends have an unreasonably low bias with respect to SHADOZ, IAGOS and OMI/MLS trends (**Table 6**), reinforcing a need for multi-decade datasets given that FT ozone changes are relatively small.

**4.2 Comparison of this study to related TOAR-II investigations**

How do our tropical tropospheric ozone trends compare to those in other studies that use SHADOZ and IAGOS profiles and/or satellite data? **Table 6** summarizes our results for the FT, UT segments and for TrCO. The FT ozone comparisons are made with Gaudel et al. (2024) and with results for two HEGIFTOM studies (HEGIFTOM-1; HEGIFTOM-2). For UT ozone, SHADOZ trends are compared to those derived from the lowest 3 layers of MLS (Froidevaux et al., 2025; see their Table 2). TrCO trends are taken from the 5 SHADOZ stations, OMI/MLS (this study) and IASI/Metop (Boynard et al., 2025). Note that the latter study only spans 2008-2023, much less than the SHADOZ data but close to the OMI/MLS period, 2005-2023.

The tropical trends study of Gaudel et al. (2024) groups SHADOZ and IAGOS profiles somewhat differently from this study and only extends through 2019 (period of trends are shown in the 4[th] column of **Table 6**). Our trends and those of Gaudel et al. (2024) for TrCO$_{sonde}$ (**Table 6, Fig. 16a**) for the equatorial Americas fall within range of one another. They fall within the wide range of the satellite products: (3.1±2.5)%/decade for OMI/MLS (2005-2023) and (-4.0±1.1)%/decade for IASI/Metop (2008-2023), an offset of 7.1%/decade for the median trends. The equatorial Americas FT ozone trends range from (-0.01%/decade to -3.3%)/decade (**Fig. 16b**).

For the Atlantic and western Africa regions, the profile-based comparisons differ in station-airport combinations among our study and analyses in Gaudel et al. (2024), HEGIFTOM-1 and HEGIFTOM-2. The TrCO and FT trends (**Table 6, Figs. 16a** and **16b**) among the 4 studies fall in a relatively small range: (-1.3±2.8)%/decade (Gaudel et al, 2024) to (1.9±2.2)%/decade (this study) for FT and (-

1.1±0.9)%/decade (Gaudel et al, 2024) to (1.9±1.8)%/decade (this study) for TrCO. The larger TrCO
from the Natal-Ascension combination appears to result from a higher positive trend in the UT
(3.4±2.9)%/decade (**Fig. 16c**). Because the FT ozone trend of Gaudel et al. (2024) is negative and the
Nat-Asc FT ozone trend is positive (**Table 6**), combining western African IAGOS profiles with the Natal
and Ascension measurements, reduces the larger area trend compared to the sonde-only FT ozone
trend. **Table 6** shows that for all regions the MLS-derived UT ozone trend estimates fall between
+3%/decade and +4%/decade (Froidevaux et al. 2025). Only over the Atlantic region do any of the
SHADOZ UT ozone trends fall in this range (**Fig. 16c**). Overall, the Atlantic and western Africa FT ozone
and $TrCO_{sonde}$ trends are 0-2%/decade, in agreement with OMI/MLS (1.3±1.4)%/decade. As for the
equatorial Americas, the 2008-2023 trends from IASI/Metop over the Atlantic and western Africa is
much lower: (-4.9±2.0)%/decade. Compared to the OMI/MLS trend, the IASI/Metop median trend is
lower by 6.2%/decade. The picture for east Africa (both SHADOZ and IAGOS profiles over Nairobi) is
similar to the Atlantic and western Africa. FT ozone, $TrCO_{sonde}$ and OMI/MLS TrCO trends are essentially
null, similar to HEGIFTOM (**Table 6, Figs. 16a** and **16b**) but IASI/Metop displays a (-3.6±1.0)%/decade
trend.

There is more variability in trends among the ground-based studies for the equatorial SE Asia region

than the equatorial Americas, Atlantic and Africa, most likely because different combinations of IAGOS
profiles and SHADOZ data were used. Supplementing FT KL-Java SHADOZ profiles with IAGOS data
(**Tables 3** and **6**) in our study, a 50% increase in sample size, did not change the trend appreciably:
(1.0±2.5)%/decade vs (1.6±3.2)%/decade. The corresponding $TrCO_{sonde}$ over KL-Java increased
(2.6±2.3)%/decade. S24 computed trends with MLR for KL-Java for 1998-2022: the $TrCO_{sonde}$ was
(3.4±2.6)%/decade. Although the trend period only differs by one year, the smaller trend with the extra
year in this study (to 2023) might reflect some COVID impact (columns 5 and 7 in **Table 2**). For
OMI/MLS TrCO, determined from a mean of changes averaged over $5^o x 5^o$ grid boxes for KL and
Watukosek (Java), there was a (5.6±6.0)%/decade, 2005-2023. The latter change is nearly the same as
Gaudel et al. (2024) for both FT ozone and OMI/MLS changes over the same interval. There are several
reasons for why Gaudel et al. (2024) FT ozone trends in **Table 6** and **Fig. 16b** are larger than our
SHADOZ-IAGOS FT ozone trends. First, the fusing of SHADOZ and IAGOS profiles in Gaudel et al. (2024)
may be more heavily weighted to polluted IAGOS segments than our merging. Second, reprocessed
IAGOS profiles have not been rigorously compared to SHADOZ data up to this point; a new evaluation of
IAGOS instrumentation in the World Calibration Center for Ozonesondes may facilitate consistent
referencing to an absolute ozone standard in the future (Smit et al., 2024; Smit et al., 2025). Third, with
the shorter trend period ending in 2019, the data of Gaudel et al. (2024) would not be affected by lower,
COVID-perturbed ozone concentrations in 2020-2023 as some SHADOZ records were (**Table 2**).

The HEGIFTOM-1 FT ozone trend (2000-2022) for Kuala Lumpur is nearly identical to the KL-Java trend, 1998-2023: (2.2±3.0)%/decade. In HEGIFTOM-2 (based on SHADOZ KL and IAGOS from several SE Asia airports) FT ozone increases over a longer period (1995-2022) are (6.2±2.2)%/decade. Thus, in general, over SE Asia, as for the equatorial Americas, Atlantic and Africa, the GB and OMI/MLS trends are in reasonable agreement, given some differences in data selection and minor differences in trend start and end dates: FT and TrCO ozone increases ~+(2-8)%/decade. Likewise, the IASI/Metop trend for TrCO, (0.0±1.4)%/decade over 2008-2023 (Boynard et al., 2025) is an outlier over SE Asia.

Nowhere are the satellite data as divergent from the FT sonde trends, near zero change from our study (1998-2023), Gaudel et al. (2024, for 2004-2019) and HEGIFTOM-1 (2000-2022), than at Samoa (**Table 6**). TrCO from the SHADOZ sondes has no significant change: (-1.4 ±4.8)%/decade, this study); (-3.1±5.4)%/decade (Gaudel et al. 2024). However, the OMI/MLS trend for TrCO is (9.1±8.3)%/decade, 2005-2023, and IASI/Metop is (-9.0±1.7)%/decade (**Fig. 16a**). The large disagreement in trends from sondes and both satellite instruments, the latter with median TrCO trends that are offset by 18% from one another, underscore the need for profile-based tropospheric ozone trends as an unbiased reference.

### 4.3  Implications of this study for TOAR II and related assessments

How do our findings apply to an overall TOAR II assessment for tropical ozone? First, there is consensus among the four GB studies: this study, Gaudel et al (2024) and the two HEGIFTOM articles (HEGIFTOM-1; HEGIFTOM-2). These results provide well-characterized trends in FT ozone for input to climate models and a reference for evaluating TrCO, total tropospheric column ozone trends from evolving tropospheric ozone satellite products. In short, over the past 20-25 years, except for equatorial SE Asia, tropical FT and total tropospheric ozone trends are negligible to within ±(1-2)%/decade (**Fig. 9**). For SE Asia both FT and TrCO have increased from ~(2-8)%/decade. Detailed analyses of sonde profiles by S24 suggest that the annually averaged FT ozone trends are (2-3)%/decade. The (5-7)%/decade increases are seasonal in FT ozone. However, for TrCO it is the large ozone increases in the boundary layer found throughout the year that are dominating the total tropospheric column change (**Fig. 11**).

Second, as far as methods of computing trends, we have shown both the similarity of median trends from QR and MLR as relative advantages of each with SHADOZ and combined SHADOZ-IAGOS profile data. HEGIFTOM-1 reached similar conclusions about QR and MLR trends for 34 ozonesonde stations, including 10 from SHADOZ. The complementarity of QR and MLR for trends attribution was also highlighted in HEGIFTOM-1. The HEGIFTOM papers (HEGIFTOM-1, HEGIFTOM-2), with a multi-instrument perspective, show variable trends among 55 global stations, including over western Europe and North America where both positive and negative tropospheric column trends occur within few

hundred km. Overall, except for southeast Asia, FT ozone trends from HEGIFTOM are small to moderate
and not distinguishable from zero at a number of tropical and extratropical locations. Third, both
HEGIFTOM-1 and this study investigated the matter of sample density in ground-based data, the former
by cutting sample size roughly in half for TrCO and our SHADOZ-IAGOS analyses for FT ozone, roughly
doubling it. In both studies trend changes determined with MLR were small (a few tenths of a
ppbv/decade) and included both increases and decreases (well-illustrated in Fig. S6 in HEGIFTOM-1).
Uncertainty changes ranged from 15-30%, usually improving with more sampling. We conclude that
arguments for excluding ground-based trends from TOAR II on the basis of sample size have little merit,
particularly given ongoing uncertainties and limited record lengths of many satellite products. Fourth,
by reducing the trend period from 26 years to 12 years, we quantified the degradation of too-short
trends for the relatively small changes that apply to much of global FT ozone.
The length of record is only one issue for ozone products derived from satellites that have operated
less than two decades. Trends based on the lowermost levels of ozone from MLS (Froidevaux et al.,
2025) were largely the same at all SHADOZ stations, (3-4%/decade for the UT, **Fig. 16c**) whereas the
sonde-derived trends are geographically variable. Only one of 5 tropical regions we analyzed, Atlantic
and western Africa, displayed a UT ozone trend in this range. The discrepancies between GB-based
trends are not surprising because MLS does not capture observed variability in UT ozone
concentrations, up to a factor of 2 between the Atlantic and eastern Indian Ocean, that is detected by the
sondes or UV sensors (Thompson et al., 2003; Thompson et al., 2017). We compared our profile-based
trends to two typical satellite products: IASI/Metop and the well-characterized OMI/MLS. Within the
uncertainty of the satellite trends, OMI/MLS TrCO trends agreed with $TrCO_{sonde}$ trends, but the largest
differences (positive trends versus negative trends respectively) were over the equatorial Americas and
Samoa. Where sonde-derived trends were near-zero, IASI/Metop trends sometimes agreed well but for
equatorial SE Asia the IASI/Metop zero trend clearly underestimated trends relative to those derived
from SHADOZ, SHADOZ-IAGOS combined profiles and OMI/MLS. At Samoa IASI/Metop (-9%/decade)
and OMI/MLS (+9%/decade) were both in error, displaying much greater disagreement with the sonde-
based trend than anywhere else. There is no easy explanation, but the differing wavelengths, vertical
sensitivities, and respective biases in the satellite products likely all play a role.
The S24 TOAR-II study is a reminder that the near-zero annually averaged FT ozone trends over most
of the tropics (**Figs. 8** and **13**) may mask strong seasonal trends (T21). S24 linked the strong February-
April increase in FT ozone over KL-Java (1998 to 2022) to declining convection with 4 proxies, e.g.,
outgoing long-wave radiation, velocity potential at 200 hPa. A role for changing dynamics must be
considered in tropical tropospheric ozone increases; increasing ozone precursor emissions are
apparently not the only driver. In a similar manner, decreases observed in LMS ozone appear to be
related to tropopause height changes during the 1998 to 2023 period.
In summary, our updated trends for SHADOZ stations (**Figs. 9** and **11**) remain the most reliable trend
reference throughout the tropical troposphere and LMS. We provide a definitive standard for evaluating
monthly trends and regional variability of satellite-based products and related models being used for
tropical trends in the past ~25 years. Together with the more extensive HEGIFTOM and IAGOS coverage,
our analyses lead to a conclusion that the most reliable trends for TOAR II are based on re-processed GB
measurements. However, coverage of the GB instruments is uneven and space-based observations are
needed for truly global trends. In order for satellite products to mature and for differences among them
to be understood, independent ozone data collection, particularly from ozonesondes referenced to an
absolute standard are required. As our investigations with the high-resolution SHADOZ profiles have
demonstrated, sonde data will remain the gold standard for deriving trends in the FT and LMS, the two
most critical regions where ozone interacts with the climate system.
**Data Availability**
All datasets used in this study are openly and publicly accessible.
V06 SHADOZ data are available at https://doi.org/10.57721/SHADOZ-V06 (NASA Goddard Space Flight Center
(GSFC) SHADOZ Team, 2019).
IAGOS and SHADOZ HEGIFTOM data are available at: https://hegiftom.meteo.be
Trends in table (*.csv) format are available at: https://tropo.gsfc.nasa.gov/shadoz/SHADOZ_PubsList
OMI/MLS data are available at https://acd-ext.gsfc.nasa.gov/Data_services/cloud_slice/new_data.html.

**Acknowledgments**
We thank Kai-Lan Chang for the QR code provided for TOAR II. Support is gratefully acknowledged from the NASA
Upper Air Research Program (K. W. Jucks, Program Manager), the SAGE III Program (R. Eckman and K. E.
Knowland, Managers), and S-NPP and JPSS (J. F. Gleason, Project Scientist). Quito monitoring was conducted from
2014 to 2021 with the support of Universidad San Francisco de Quito USFQ and the Vienna Convention Trust
Fund.
**Author Contributions**
Conceptualization of the study was by AMT, RMS, DEK. AMT led the writing with RMS and DEK, additional input
from GAM and KM. Analyses were carried out by RMS and DEK. All authors collected and/or contributed data.
**Editorial Input.** The article was handled by Editor E. Landulfo and benefitted from two anonymous reviewers.
**Competing Interests.** All authors declare that we have no competing interests.
**Acronyms and Chemical Names**
AIRS = Atmospheric Infrared Sounder
CAM-Chem = Community Atmosphere Model with Chemistry
$CH_4$ = Methane
DU = Dobson Units
ECC = Electrochemical Concentration Cell
ENSO = El Niño Southern Oscillation
FT = Free troposphere
FTIR = Fourier Transform Infrared
GAW = Global Atmospheric Watch
GB = Ground-based
HEGIFTOM = Harmonization and Evaluation of Ground-based Instruments for Free-Tropospheric Ozone
Measurements
IASI = Infrared Atmospheric Sounding Interferometer
IGAC = International Global Atmospheric Chemistry
IOD = Indian Ocean Dipole
ITCZ = Inter-Tropical Convergence Zone
KL = Kuala Lumpur
LMS = Lowermost stratosphere
MEI = Multivariate ENSO Index
MLR = Multi-linear Regression
MLS = Microwave Limb Sounder
NDACC = Network for the Detection of Atmospheric Composition Change
$O_3$ = Ozone
OH = Hydroxyl (radical)
OMI = Ozone Monitoring Instrument
QBO = Quasi-biennial Oscillation
QR = Quantile Regression
SC-Para = San Cristóbal-Paramaribo
SHADOZ = Southern Hemisphere Additional Ozonesondes
TOAR = Tropospheric Ozone Assessment Report
TH = Tropopause Height
TCO = Total Column Ozone
TrCO = Tropospheric Column Ozone (TrOC in HEGIFTOM)
TROPESS = TRopospheric Ozone and its Precursors from Earth System Sounding
UV = Ultraviolet
WAACM = Whole Atmosphere Community Climate Model
WMO = World Meteorological Organization

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

## Tables and Figures

**Table 1.** List of the 27 total SHADOZ and IAGOS sites, and their metadata, used in this analysis.

| Site | Country | Observation Network | Latitude | Longitude | Altitude (m) |
|---|---|---|---|---|---|
| Abidjan (ABJ) | Cote d'Ivoire | IAGOS | 5.25 | -3.93 | 6 |
| Accra (ACC) | Ghana | IAGOS | 5.61 | -0.17 | 62 |
| Addis Ababa (ADD) | Ethiopia | IAGOS | 8.98 | 38.80 | 2326 |
| *Ascension Island* | United Kingdom | SHADOZ | -7.58 | -14.24 | 85 |
| Bogota (BOG) | Colombia | IAGOS | 4.70 | -74.14 | 2548 |
| Brazzaville (BZV) | Congo (Brazzaville) | IAGOS | -4.26 | 15.25 | 319 |
| Caracas (CCS) | Venezuela | IAGOS | 10.60 | -67.01 | 71 |
| Cotonou (COO) | Benin | IAGOS | 6.35 | 2.39 | 6 |
| Douala (DLA) | Cameroon | IAGOS | 4.01 | 9.72 | 10 |
| *Hanoi* | Vietnam | SHADOZ | 21.01 | 105.80 | 6 |
| *Hilo, Hawaii* | United States | SHADOZ | 19.43 | -155.04 | 11 |
| *Irene* | South Africa | SHADOZ | -25.90 | 28.22 | 1524 |
| Kinshasa (FIH) | Congo (Kinshasa) | IAGOS | -4.39 | 15.45 | 313 |
| *Koror* | Palau | SHADOZ | 7.34 | 134.47 | 23 |
| *Kuala Lumpur* | Malaysia | SHADOZ | 2.73 | 101.27 | 17 |
| Kuala Lumpur (KUL) | Malaysia | IAGOS | 2.76 | 101.71 | 21 |
| Lagos (LOS) | Nigeria | IAGOS | 6.58 | 3.32 | 41 |
| Libreville (LBV) | Gabon | IAGOS | 0.46 | 9.41 | 12 |
| Lome (LFW) | Togo | IAGOS | 6.17 | 1.25 | 22 |
| Luanda (LAD) | Angola | IAGOS | -8.85 | 13.23 | 74 |
| Malabo (SSG) | Equatorial Guinea | IAGOS | 3.76 | 8.72 | 23 |
| *Nairobi* | Kenya | SHADOZ | -1.27 | 36.80 | 1795 |
| *Natal* | Brazil | SHADOZ | -5.42 | -35.38 | 42 |
| *Pago Pago* | American Samoa | SHADOZ | -14.23 | -170.56 | 77 |
| *Paramaribo* | Surinam | SHADOZ | 5.80 | -55.21 | 23 |
| Port Harcourt (PHC) | Nigeria | IAGOS | 5.01 | 6.95 | 27 |
| *Quito* | Ecuador | SHADOZ | -0.20 | -78.44 | 2414 |
| *Reunion Island* | France | SHADOZ | -21.06 | 55.48 | 10 |
| *San Cristobal* | Ecuador | SHADOZ | -0.89 | -89.61 | 8 |
| Sao Paulo (GRU) | Brazil | IAGOS | -23.43 | -46.48 | 750 |
| Singapore (SIN) | Singapore | IAGOS | 1.36 | 103.99 | 7 |
| *Suva* | Fiji | SHADOZ | -18.13 | 178.40 | 6 |
| *Watukosek, Java* | Indonesia | SHADOZ | -7.46 | 112.43 | 50 |
| Yaounde (NSI) | Cameroon | IAGOS | 3.70 | 11.55 | 694 |

**Table 2.** SHADOZ metadata: number of profiles, annual trends. Each row indicates a different segment: 5-10km, 10-15km, 15-20km, TH-5km to TH, TH to TH +5km, and surface to Tp (tropopause). Periods analyzed (columns) are 1998-2019 (T21), 1998-2023, 2000-2023; 2005-2023 for OMI/MLR comparisons in total tropospheric ozone column amount (TrCO). Annually-averaged MLR partial column ozone linear trends are shown DU per decade and in percent per decade, with the 95% confidence interval. Trends with p-values <0.05 are shown in bold.

| SHADOZ T21 Updated MLR FT Ozone Trends | | | | | | | | | | |
|---|---|---|---|---|---|---|---|---|---|---|
| Station | Altitude Range | Number of Profiles | 1998-2019 T21 Annual Trend ± 2*sigma (DU/decade) | 1998-2019 T21 Annual Trend ± 2*sigma (%/decade) | 1998-2023 Annual Trend ± 2*sigma (DU/decade) | 1998-2023 Annual Trend ± 2*sigma (%/decade) | 2000-2023 Annual Trend ± 2*sigma (DU/decade) | 2000-2023 Annual Trend ± 2*sigma (%/decade) | 2005-2023 Annual Trend ± 2*sigma (DU/decade) | 2005-2023 Annual Trend ± 2*sigma (%/decade) |
| San Cristobal – Paramaribo | 5-10km | 1370 | 0.2±0.3 | 1.9±3.1 | -0.1±0.3 | -0.9±3.7 | -0.1±0.3 | -1.2±3.8 | -0.3±0.5 | -3.6±5.1 |
| | 10-15km | | 0.1±0.2 | 1.5±4.0 | -0.2±0.3 | -3.5±4.5 | -0.2±0.3 | -4.0±4.4 | **-0.4±0.4** | **-6.6±6.5** |
| | 15-20km | | **-0.4±0.4** | **-3.1±2.8** | **-0.5±0.4** | **-3.9±2.9** | -0.4±0.4 | -2.8±3.2 | -0.4±0.7 | -3.3±5.5 |
| | TH-5km to TH | | 0.0±0.2 | 0.2±4.2 | -0.2±0.3 | -2.9±5.1 | -0.2±0.3 | -2.9±5.2 | -0.3±0.4 | -5.9±7.5 |
| | TH to TH+5km | | 0.2±0.6 | 0.6±2.3 | -0.3±0.7 | -1.2±2.6 | -0.3±0.8 | -1.1±2.9 | -0.4±1.3 | -1.5±4.5 |
| | TrCO, surf-Tp | | NA | NA | -0.3±0.9 | -1.0±3.3 | -0.2±1.0 | -0.8±3.6 | -0.4±1.6 | -1.4±5.9 |
| Natal - Ascension Island | 5-10km | 1646 | 0.2±0.3 | 1.6±2.3 | 0.2±0.3 | 1.9±2.2 | 0.1±0.2 | 0.5±1.8 | 0.0±0.4 | 0.3±2.9 |
| | 10-15km | | **0.3±0.2** | **3.9±2.8** | **0.2±0.2** | **3.4±2.9** | 0.1±0.2 | 1.7±2.4 | 0.1±0.3 | 0.7±3.8 |
| | 15-20km | | -0.0±0.3 | -0.4±2.4 | -0.1±0.3 | -1.0±2.3 | -0.2±0.3 | -1.4±2.6 | -0.3±0.5 | -2.4±3.8 |
| | TH-5km to TH | | **0.3±0.2** | **4.7±2.7** | **0.2±0.2** | **3.4±2.9** | 0.1±0.2 | 1.7±2.4 | 0.0±0.2 | 0.2±3.3 |
| | TH to TH+5km | | 0.5±0.5 | 1.9±1.9 | 0.2±0.7 | 0.9±2.7 | -0.0±0.6 | -0.1±2.5 | -0.4±0.9 | -1.6±3.7 |
| | TrCO surf-Tp | | NA | NA | **0.7±0.6** | **1.9±1.8** | 0.3±0.7 | 0.9±1.9 | 0.3±1.0 | 1.0±2.8 |
| Nairobi | 5-10km | 976 | 0.1±0.3 | 1.2±3.1 | 0.1±0.3 | 0.5±3.0 | 0.1±0.4 | 1.0±3.5 | -0.0±0.7 | -0.3±6.3 |
| | 10-15km | | -0.0±0.2 | -0.2±3.4 | -0.1±0.2 | -1.5±3.2 | -0.1±0.3 | -1.9±4.2 | -0.2±0.6 | -2.4±8.2 |
| | 15-20km | | 0.1±0.3 | 0.6±2.5 | 0.1±0.5 | 0.9±3.9 | 0.3±0.5 | 2.4±4.2 | 0.7±0.9 | 5.6±6.9 |
| | TH-5km to TH | | 0.0±0.2 | 0.7±3.2 | -0.0±0.2 | -0.0±2.5 | -0.0±0.2 | -0.1±3.3 | -0.0±0.4 | -0.2±6.1 |
| | TH to TH+5km | | 0.5±0.7 | 1.9±2.7 | 0.4±0.9 | 1.4±3.5 | 0.5±1.1 | 1.7±4.2 | 1.2±1.7 | 4.5±6.3 |
| | TrCO, Surf-Tp | | NA | NA | 0.3±0.7 | 1.1±2.5 | 0.3±0.9 | 1.0±3.2 | -0.4±1.5 | -1.5±5.2 |
| Kuala Lumpur – Watukosek | 5-10km | 870 | 0.1±0.2 | 1.9±3.0 | 0.1±0.2 | 1.0±2.5 | 0.1±0.2 | 1.0±3.1 | -0.1±0.3 | -1.2±4.3 |
| | 10-15km | | -0.0±0.1 | -0.6±3.3 | 0.0±0.1 | 1.3±3.6 | 0.1±0.2 | 2.9±4.2 | 0.2±0.2 | 4.3±6.6 |
| | 15-20km | | **-0.7±0.3** | **-5.8±2.8** | -0.3±0.6 | -2.4±4.8 | -0.1±0.6 | -0.4±5.3 | 0.6±0.8 | 5.2±6.8 |
| | TH-5km to TH | | -0.1±0.1 | -3.2±3.3 | 0.0±0.2 | 0.8±5.7 | 0.1±0.2 | 2.6±6.8 | 0.2±0.3 | 5.1±8.8 |

| | | | | | | | | | |
|---|---|---|---|---|---|---|---|---|---|
| | TH to TH+5km | | -0.1±0.8 | -0.5±3.0 | 0.2±1.1 | 0.9±4.2 | 0.3±1.2 | 1.1±4.5 | **1.7±1.0** | **7.0±4.0** |
| | TrCO, surf-Tp | | NA | NA | **0.6±0.6** | **2.6±2.3** | **1.1±0.7** | **4.6±2.8** | 0.7±1.1 | 3.0±4.6 |
| Samoa | 5-10km | 928 | 0.1±0.3 | 1.4±4.7 | 0.1±0.3 | 0.8±4.4 | -0.0±0.3 | -0.2±4.5 | -0.2±0.4 | -3.0±5.7 |
| | 10-15km | | 0.1±0.3 | 2.5±6.5 | -0.0±0.4 | -1.3±9.2 | -0.1±0.4 | -3.0±9.4 | **-0.4±0.4** | **-10.0±10.0** |
| | 15-20km | | -0.4±0.5 | -2.8±3.4 | -0.3±0.7 | -2.3±5.2 | -0.4±0.7 | -2.9±5.3 | **-1.0±0.8** | **-7.0±5.4** |
| | TH-5km to TH | | 0.0±0.3 | 0.2±6.5 | -0.1±0.4 | -1.7±8.8 | -0.1±0.4 | -2.5±9.5 | **-0.5±0.4** | **-10.6±9.0** |
| | TH to TH+5km | | -0.3±0.7 | -0.9±2.4 | -0.4±0.9 | -1.4±3.1 | -0.9±0.9 | -2.9±3.0 | **-1.2±1.1** | **-3.9±3.7** |
| | TrCO, surf-Tp | | NA | NA | -0.3±1.0 | -1.4±4.8 | -0.3±1.1 | -1.3±5.4 | -0.9±1.4 | -4.4±6.5 |

1088

**Table 3.** SHADOZ and IAGOS combined MLR ozone trends values for FTp (700-300 hPa) partial column for 5 regions: Equatorial Americas, Atlantic and West Africa, East Africa, Equatorial Southeast Asia, and Samoa (individual record). The individual sites are listed for each region. Annually-averaged MLR partial column ozone linear trends are shown DU per decade and in percent per decade, with the 95% confidence interval. Trends with p-values <0.05 are shown in bold.

| SHADOZ MLR Regional FT (700-300 hPa) Ozone Trends | | | | |
|---|---|---|---|---|
| **Region Name** | **Individual SHADOZ & IAGOS Locations (IAGOS regions in bold)** | **Number of Profiles** | **1998-2023 Annual Trend ± 2*sigma (DU/decade)** | **1998-2023 Annual Trend ± 2*sigma (%/decade)** |
| Equatorial Americas | San Cristobal (Ecuador), Paramaribo (Suriname), Quito (Ecuador), Caracas (Venezuela), Bogota (Colombia) | 2821 | 0.00 ± 0.31 | -0.01 ± 2.57 |
| Atlantic and West Africa | Natal (Brazil), Ascension Island (UK); **Central Africa** [Luanda (Angola), Brazzaville (Congo), Kinshasa (Democratic Republic of Congo)], **Gulf of Guinea** [ Lomé (Togo), Yaoundé (Cameroon), Douala (Cameroon), Libreville (Gabon), Accra (Ghana), Abidjan (Ivory Coast), Malabo (Equatorial Guinea), Cotonou (Benin), Port Harcourt (Nigeria)], **Lagos** (Nigeria) | 4271 | 0.12 ± 0.39 | 0.69 ± 2.28 |
| East Africa | Nairobi (Kenya), Addis Ababa (Ethiopia) | 1297 | 0.12 ± 0.38 | 0.85 ± 2.69 |
| Equatorial Southeast Asia | Kuala Lumpur (Malaysia), Watukosek (Indonesia); **Gulf of Thailand** [Kuala Lumpur (Malaysia), Singapore (Singapore)] | 1305 | 0.16 ± 0.34 | 1.57 ± 3.25 |
| Samoa | Pago Pago (Am. Samoa) | 928 | -0.04 ± 0.38 | -0.42 ± 3.85 |

**Table 4.** SHADOZ QR median (50%-ile) annual ozone trends values (1998-2023) for TrCO (surface to tropopause) and FTp (700-300 hPa) columns in DU/decade and in %/decade with ±2*sigma. Trends

 with p-values <0.05 are shown in bold. L1 data is daily data from the HEGIFTOM database
(https://hegiftom.meteo.be/datasets).

| SHADOZ QR 1998-2023 L1 Annual Ozone Trends | | | | | | | |
|---|---|---|---|---|---|---|---|
| Station | Latitude | Longitude | L1 Obs # | TrCO (surf-Tp) Trend ± 2*sigma (DU/decade) | TrCO (surf-Tp) Trend ± 2*sigma (%/decade) | FTp (300-700 hPa) Trend ± 2*sigma (DU/decade) | FTp (300-700 hPa) Trend ± 2*sigma (%/decade) |
| **Samoa** | -14.23 | -170.56 | 797 | -0.29 ± 0.54 | -1.41 ± 2.59 | -0.04 ±0.36 | -0.44 ± 3.58 |
| *Hilo* | *19.43* | *-155.04* | *1142* | *-0.71 ± 0.72* | *-2.21 ± 2.25* | *-0.20 ± 0.24* | *-1.33 ± 1.51* |
| **San Cristobal** | -0.89 | -89.61 | 350 | -0.44 ± 1.14 | -1.69 ± 4.42 | -0.03 ±0.48 | -0.24 ± 3.79 |
| **Paramaribo** | 5.80 | -55.21 | 855 | 0.22 ± 0.62 | 0.80 ± 2.27 | 0.09 ± 0.34 | 0.67 ± 2.62 |
| **Natal** | -5.42 | -35.38 | 676 | **1.04 ± 0.68** | **3.04 ± 2.00** | **0.56 ± 0.48** | **3.29 ± 2.79** |
| **Ascension Island** | -7.58 | -14.24 | 676 | 0.20 ± 0.84 | 0.53 ± 2.19 | 0.04 ± 0.40 | 0.21 ± 2.09 |
| *Irene* | *-25.90* | *28.22* | *387* | *0.53 ± 0.98* | *1.44 ± 2.63* | *0.38 ± 0.48* | *2.02 ± 2.63* |
| **Nairobi** | -1.27 | 36.80 | 872 | 0.26 ± 0.70 | 0.94 ± 2.53 | 0.16 ± 0.46 | 1.11 ± 3.13 |
| *Reunion* | *-21.06* | *55.48* | *735* | ***2.63 ± 0.78*** | ***7.22 ± 2.14*** | ***0.90 ± 0.40*** | ***5.22 ± 2.31*** |
| **Kuala Lumpur** | 2.73 | 101.27 | 456 | 0.67 ± 1.12 | 2.68 ± 3.69 | 0.10 ± 0.50 | 0.95 ± 4.77 |
| *Hanoi\** | *21.01* | *105.80* | *350* | *1.94 ± 2.18* | *4.85 ± 5.45* | *0.43 ± 0.96* | *2.46 ± 5.52* |
| **Watukosek** | -7.46 | 112.43 | 326 | 0.71 ± 1.74 | 2.87 ± 6.99 | 0.22 ± 0.78 | 2.07 ± 7.26 |
| **Fiji** | -18.13 | 178.40 | 391 | -0.28 ± 0.84 | -1.16 ± 3.54 | 0.17 ± 0.68 | 1.48 ± 5.71 |

* Hanoi dataset starts in 2004 (not 1998).

**Table 5.** SHADOZ QR median (50%-ile) annual ozone trends values (1998-2023 and 2008-2019) for TrCO (surface to tropopause) columns in DU/decade with ±2*sigma. Trends with p-values <0.05 are

shown in bold. L1 data is daily data from the HEGIFTOM database
(https://hegiftom.meteo.be/datasets).

| SHADOZ QR L1 Annual TrCO Trends for 1998-2023 and 2008-2019 | | | | |
|---|---|---|---|---|
| Station | Latitude | Longitude | 1998-2023 TrCO (surf-Tp) Trend ± 2*sigma (DU/decade) | 2008-2019 TrCO (surf-Tp) Trend ± 2*sigma (DU/decade) |
| Samoa | -14.23 | -170.56 | -0.29 ± 0.54 | **-1.74 ± 1.70** |
| Hilo | 19.43 | -155.04 | **-0.71 ± 0.72** | **-1.46 ± 1.46** |
| San Cristobal | -0.89 | -89.61 | -0.44 ± 1.14 | -0.39 ± 2.42 |
| Paramaribo | 5.80 | -55.21 | 0.22 ± 0.62 | 0.47 ± 1.26 |
| Natal | -5.42 | -35.38 | **1.04 ± 0.68** | -1.06 ± 2.06 |
| Ascension Island | -7.58 | -14.24 | 0.20 ± 0.84 | 1.26 ± 2.04 |
| Irene | -25.90 | 28.22 | 0.53 ± 0.98 | -1.76 ± 2.74 |
| Nairobi | -1.27 | 36.80 | 0.26 ± 0.70 | 0.39 ± 1.58 |
| Reunion | -21.06 | 55.48 | **2.63 ± 0.78** | 1.67 ± 1.92 |
| Kuala Lumpur | 2.73 | 101.27 | 0.67 ± 1.12 | 0.90 ± 1.86 |
| Hanoi* | 21.01 | 105.80 | 1.94 ± 2.18 | 0.80 ± 1.80 |
| Watukosek | -7.46 | 112.43 | 0.71 ± 1.74 | 0.00 ± 2.28 |
| Fiji | -18.13 | 178.40 | -0.28 ± 0.84 | -1.58 ± 2.46 |

* Hanoi dataset starts in 2004 (not 1998).

**Table 6.** Summary of annual tropopspheric (FT, UT and TrCO) ozone trends values for the tropics in ppbv/decade and %/decade with ±2*sigma (estimated in some cases where different units published) from TOAR-II relevant papers separated into 5 regions: Equatorial Americas, Atlantic and West Africa, East Africa, Southeast Asia and Samoa.  There are 6 different references, including this work,

representing ground-based and satellite observations covering the 1995-2023 time period (time ranges do vary study to study).

| Region | Reference | Data Description | Time Period | FT Trends** | | UT Trends** | | TrCO Trends** | |
|---|---|---|---|---|---|---|---|---|---|
| | | | | ppbv /decade[t] | % /decade | ppbv /decade[t] | % /decade | ppbv /decade[t] | % /decade |
| Equatorial Americas | This work | ozonesonde data from SHADOZ: San Cristobal and Paramaribo (5-10km for FT, 10-15km for UT, surface to tropopause for TrCO) | 1998-2023 | -0.6 ± 1.8 | -0.9 ± 3.7 | -2.1 ± 3.1 | -3.5 ± 4.5 | -0.5 ± 1.6 | -1.0 ± 3.3 |
| | | IAGOS and SHADOZ data for Equatorial Americas (700-300hPa for FT) | 1998-2023 | 0.00 ± 1.2 | -0.01 ± 2.6 | NA | NA | NA | NA |
| | | OMI/MLS[$] tropospheric column (surface to tropopause) | 2005-2023 | NA | NA | NA | NA | 1.4 ± 1.1 | 3.1 ± 2.5 |
| | Gaudel et al (2024) | Fused IAGOS + SHADOZ (FT from Figure S24 and TrOC from Table 1) | 2004–2019 | -1.5 ± 0.75 | -3.3 ± 1.7 | NA | NA | -1.3 ± 0.4 | -2.9 ± 0.9 |
| | Froidevaux et al (2025) | Aura MLS UT data averaged over 147 and 215 hPa based on Figure 5 and Table S1 | 2005–2020 | NA | NA | 2.0 ± 1.3 | 4.0 ± 2.7 | NA | NA |
| | Van Malderen et al (2025a) | HEGIFTOM* Paramaribo ozonesonde data (700-300 hPa for FT) from Table 2 | 2000-2022 | 0.3 ± 1.1 | 0.7 ± 3.0 | NA | NA | NA | NA |
| | Boynard et al (2025) | Climate Data Record (CDR) from IASI for TrCO (surface to thermal tropopause) based on Figures 12 & 13 | 2008-2023 | NA | NA | NA | NA | -1.8 ± 0.5 | -4.0 ± 1.1 |
| Atlantic and West Africa | This work | ozonesonde data from SHADOZ: Natal and Ascension (vertical columns same as above) | 1998-2023 | 1.2 ± 1.8 | 1.9 ± 2.2 | 2.1 ± 2.1 | 3.4 ± 2.9 | 1.3 ± 1.1 | 1.9 ± 1.8 |
| | | IAGOS and SHADOZ data for Atlantic and West Africa | 1998-2023 | 0.5 ± 1.5 | 0.7 ± 2.3 | NA | NA | NA | NA |

| Region | Reference | Description | Years | | | | | | |
|---|---|---|---|---|---|---|---|---|---|
| | | (700-300hPa for FT) | | | | | | | |
| | | OMI/MLS[$] tropospheric column (surface to tropopause) | 2005-2023 | NA | NA | NA | NA | 0.7 ± 0.9 | 1.3 ± 1.4 |
| | Gaudel et al (2024) | Natal/Ascension (SHADOZ only) and West Africa (IAGOS only); FT from Figure S24 and TrOC from Table 1 | 2004–2019 | NA -0.7 ± 1.5 | NA -1.3 ± 2.8 | NA | NA | -0.6 ± 0.5 0.4 ± 1.0 | -1.1 ± 0.9 0.7 ± 1.8 |
| | Froidevaux et al (2025) | same as above | 2005–2020 | NA | NA | 1.9 ± 1.3 | 3.7 ± 2.7 | NA | NA |
| | Van Malderen et al (2025a;2025b) | HEGIFTOM* Ascension/Natal ozonesonde data from Table 2 (ref a) and merged Gulf of Guinea data (700-300 hPa for FT) Table S3 (ref b) | 2000-2022 1995-2022 | 0.0 ± 1.5 -0.5 ± 0.7 | 0.0 ± 2.7 -1.0 ± 1.3 | NA | NA | NA | NA |
| | Boynard et al (2025) | CDR from IASI/Metop for TRCO (surface to thermal tropopause) based on Figures 12 & 13 | 2008-2023 | NA | NA | NA | NA | -2.7 ± 0.5 | -4.9 ± 2.0 |
| East Africa | This work | ozonesonde data from SHADOZ: Nairobi (vertical columns same as above) | 1998-2023 | 0.6 ± 1.8 | 0.5 ± 3.0 | -1.0 ± 2.1 | -1.5 ± 3.2 | 0.5 ± 1.3 | 1.1 ± 2.5 |
| | | IAGOS and SHADOZ data for East Africa (700-300hPa for FT) | 1998-2023 | 0.5 ± 1.5 | 0.8 ± 2.7 | NA | NA | NA | NA |
| | | OMI/MLS[$] tropospheric column (surface to tropopause) | 2005-2023 | NA | NA | NA | NA | 0.2 ± 2.0 | 0.6 ± 4.5 |
| | Gaudel et al (2024) | NA | NA | NA | NA | NA | NA | NA | NA |
| | Froidevaux et al (2025) | same as above | 2005–2020 | NA | NA | 1.8 ± 1.3 | 3.7 ± 2.7 | NA | NA |
| | Van Malderen et al (2025a) | HEGIFTOM* Nairobi ozonesonde data (700-300 hPa for FT) from Table 2 | 2000-2022 | 0.3 ± 1.5 | 0.7 ± 3.4 | NA | NA | NA | NA |
| | Boynard et al (2025) | CDR from IASI/Metop for TRCO (surface to thermal | 2008-2023 | NA | NA | NA | NA | -1.8 ± 0.5 | -3.6 ± 1.0 |

| Region | Reference | Description | Period | | | | | | |
|---|---|---|---|---|---|---|---|---|---|
| | | tropopause) based on Figures 12 & 13 | | | | | | | |
| Southeast Asia | This work | ozonesonde data from SHADOZ: Kuala Lumpur and Watukosek (vertical columns same as above) | 1998-2023 | 0.6 ± 1.2 | 1.0 ± 2.5 | 0.0 ± 1.0 | 1.3 ± 3.6 | 1.1 ± 1.1 | 2.6 ± 2.3 |
| | | IAGOS and SHADOZ data for Equatorial Southeast Asia (700-300hPa for FT) | 1998-2023 | 0.6 ± 1.3 | 1.6 ± 3.2 | NA | NA | NA | NA |
| | | OMI/MLS$ tropospheric column (surface to tropopause) | 2005-2023 | NA | NA | NA | NA | 2.0 ± 2.2 | 5.6 ± 6.0 |
| | Gaudel et al (2024) | Fused IAGOS + SHADOZ for SE Asia and Malaysia/Indonesia ; FT from Figure S24 and TrOC from Table 1 | 2004–2019 | 3.0 ± 1.5 1.5 ± 1.5 | 6.7 ± 3.3 5.0 ± 5.0 | NA | NA | 2.8 ± 1.4 3.4 ± 1.3 | 5.6 ± 2.8 8.5 ± 3.2 |
| | Stauffer et al (2024) | ozonesonde data from SHADOZ: Kuala Lumpur and Watukosek (5-10km for FT, 10-15km for UT, surface to tropopause for TrCO) | 1998-2022 | 0.5 ±1.0 | 1.3 ± 2.7 | 0.0 ± 1.0 | 1.3 ± 2.7 | 1.3 ± 1.0 | 3.4 ± 2.6 |
| | Froidevaux et al (2025) | same as above | 2005–2020 | NA | NA | 2.2 ± 1.6 (up to 8.0) | 4.0 ± 2.8 (up to 14) | NA | NA |
| | Van Malderen et al (2025a;2025b) | HEGIFTOM* Kuala Lumpur alone from Table 2 (ref a) and merged SE Asia/ Malaysia data (700-300 hPa for FT) from Table S3 (ref b) | 2000-2022 1995-2022 | 0.8 ± 1.1 3.1 ± 1.1 | 2.2 ± 3.0 6.2 ± 2.2 | NA | NA | NA | NA |
| | Boynard et al (2025) | CDR from IASI/Metop for TRCO (surface to thermal tropopause) based on Figures 12 &13 | 2008-2023 | NA | NA | NA | NA | 0.0 ± 0.5 | 0.0 ± 1.4 |
| Samoa | This work | ozonesonde data from SHADOZ: Samoa (vertical columns same as above) | 1998-2023 | 0.6 ± 1.8 | 0.8 ± 4.4 | 0.0 ± 4.1 | -1.3 ± 9.2 | -0.5 ± 1.8 | -1.4±4.8 |

| | | | | | | | | |
|---|---|---|---|---|---|---|---|---|
| | IAGOS and SHADOZ data for Samoa (700-300hPa for FT) | 1998-2023 | -0.15 ± 1.5 | -0.4 ± 3.8 | NA | NA | NA | NA |
| | OMI/MLS[$] tropospheric column (surface to tropopause) | 2005-2023 | NA | NA | NA | NA | 2.5 ± 2.3 | 9.1 ± 8.3 |
| **Gaudel et al (2024)** | Samoa (SHADOZ only) ;TrOC from Table 1 | 2004–2019 | NA | NA | NA | NA | -1.1 ± 1.9 | -3.1 ± 5.4 |
| **Froidevaux et al (2025)** | same as above | 2005–2020 | NA | NA | 1.9 ± 1.5 | 3.4 ± 2.6 | NA | NA |
| **Van Malderen et al (2025a)** | HEGIFTOM* Samoa ozonesonde data (700-300 hPa for FT) from Table 2 | 2000-2022 | -0.07 ± 1.3 | -0.2 ± 4.2 | NA | NA | NA | NA |
| **Boynard et al (2025)** | CDR from IASI/Metop for TRCO (surface to thermal tropopause) based on Figures 12 & 13 | 2008-2023 | NA | NA | NA | NA | -2.7 ± 0.5 | -9.0 ± 1.7 |

* Note HEGIFTOM data can be from the following ground networks: IAGOS, ozonesondes, FTIR, Umkehr,
and Lidar.
** Annual average trend plus 2*sigma.
[t] SHADOZ Trends have been converted from DU/decade to ppbv/decade for each layer so these are
approximates values based on Tables 1 and 2.
[$] OMI/MLS trend values are from the 5x5 deg box that contains the SHADOZ station. In the cases where
there are two SHADOZ stations (Nat/Asc, KL/Java, SC/Para), it's the average of the two OMI/MLS trends
and confidence interval values.


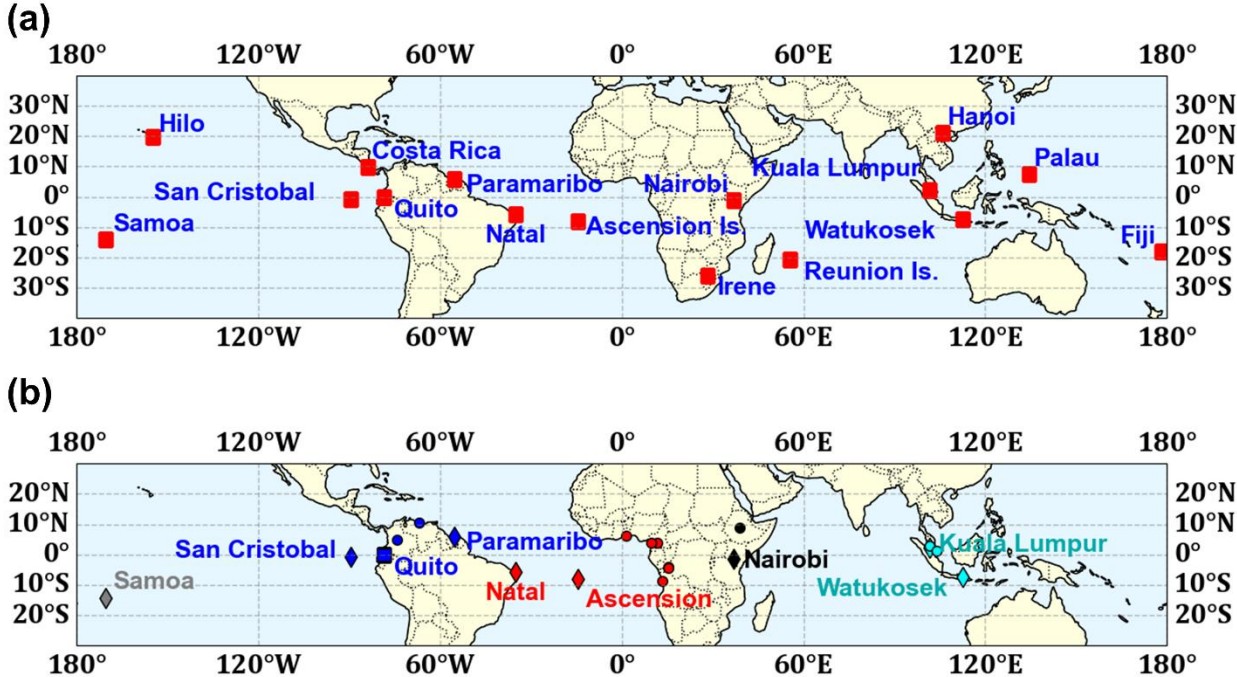

**Figure 1.** (a) Map of all SHADOZ stations in 2025 when the Quito and Palau records, that began in 2014
and 2015, respectively, joined the network. (b) SHADOZ sites used in T21 for 1998-2019 equatorial
trends and here for 1998-2023 trends are represented with diamonds. SHADOZ stations whose
combined records are examined within regions are colored blue for Equatorial Americas, red for
Atlantic and cyan for equatorial southeast Asia. The airport locations from IAGOS include are those
indicated with circles: red for Atlantic and West Africa, black for East Africa, cyan for the equatorial
Southeast Asia region. For T21 and the trends here, the fifth station, including the combined stations, is
Samoa (in gray). Quito (blue square) data are included in the Equatorial Americas trends. All station
and airport locations are in **Table 1.** The 5 SHADOZ sites (2 individual and 3 combined) from T21 and
for which trends are updated here appear in **Table 2** with the MLR trends. Individual SHADOZ and
IAGOS site names within each region and their trends are shown in **Table 3**. Trends computed by QR
for total column and FT ozone at 13 individual SHADOZ stations are summarized in **Table 4**.

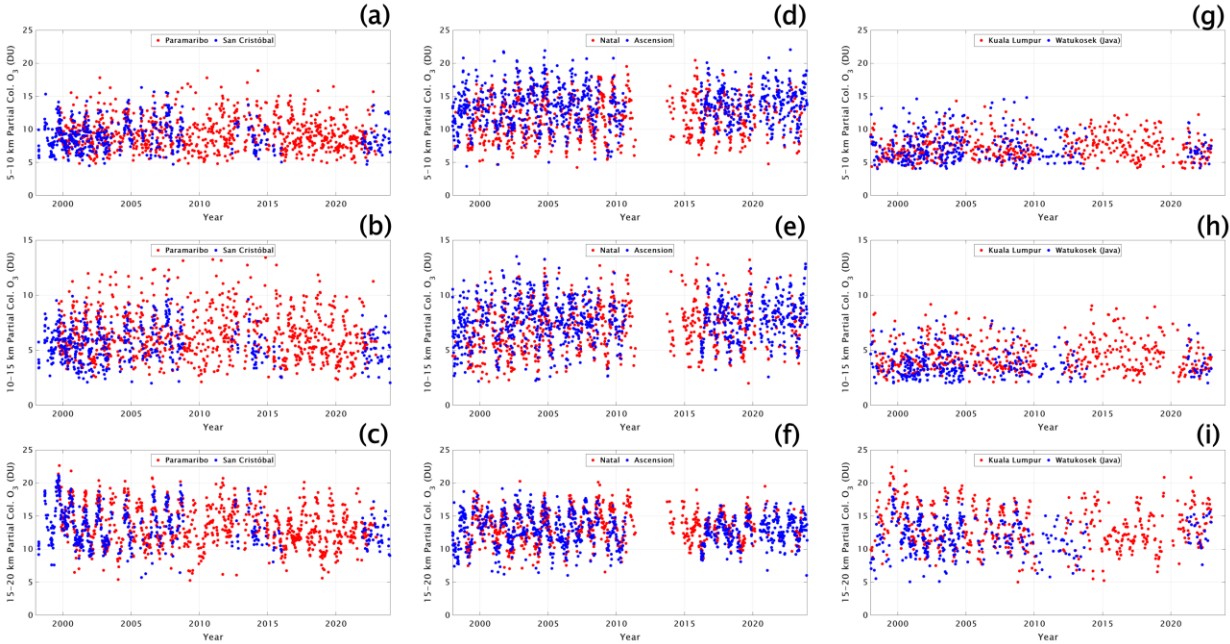

**Figure 2.** Time-series of ozone column segments (in DU) for the combined SHADOZ stations, for the
layers 5-10 km, 10-15 km, 15-20 km for: San Cristóbal-Paramaribo (a-c); Natal-Ascension (d-f); Kuala
Lumpur-Watukosek (Java) (g-i). Station coordinates in **Table 1.**

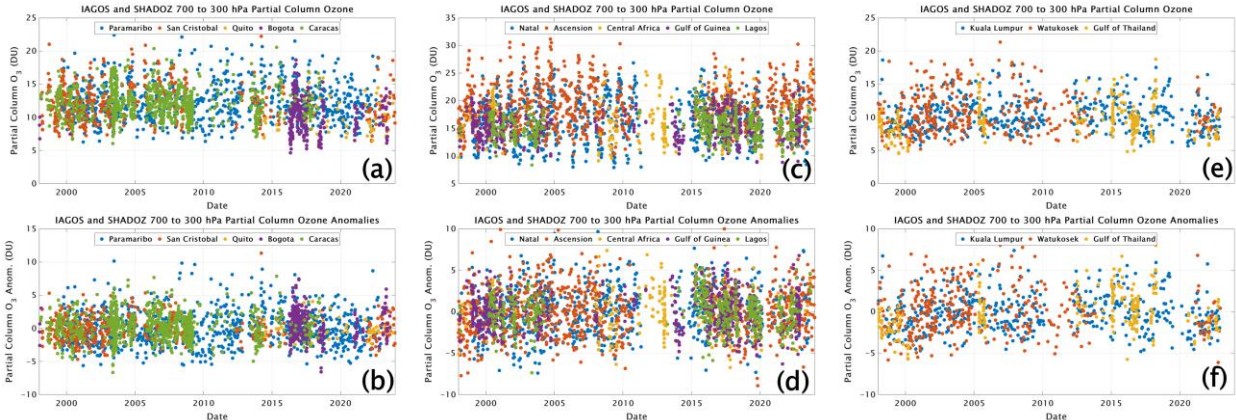

**Figure 3.** Time-series of SHADOZ and IAGOS partial ozone column amounts and partial ozone column
anomalies (in DU) for the pressure-defined mid-free troposphere, FTp (700 to 300 hPa), for (a-b)
Equatorial Americas, (c-d) Atlantic and West Africa, and (e-f) Equatorial SE Asia.  The listing of the
individual sites included in these datasets appears in **Table 3**.  Coordinates are in **Table 1.**

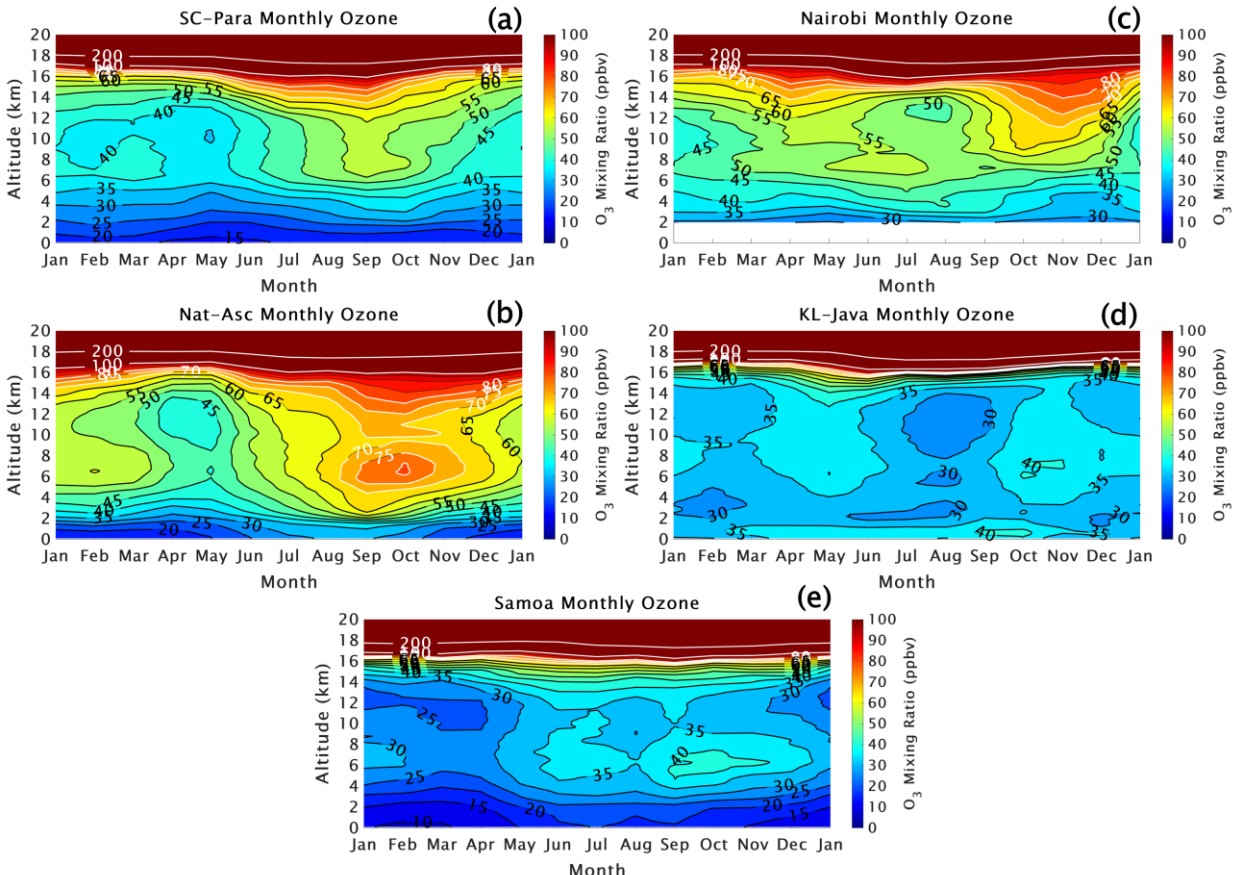

**Figure 4.** Monthly averaged ozone mixing ratios from the surface to 20 km altitude for the five SHADOZ sites: (a) San Cristóbal – Paramaribo, (b) Natal – Ascension Island, (c) Nairobi, (d) Kuala Lumpur – Watukosek (Java), and (e) Samoa.  Colors with black and white contour lines are shown for the ozone mixing ratios in ppbv.

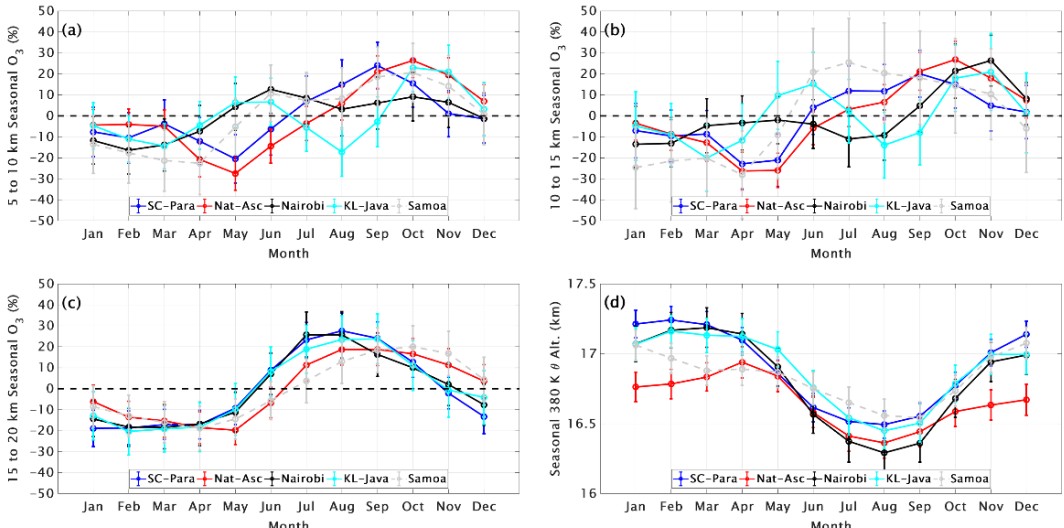

**Figure 5**.  Monthly ozone variability for the five T21 SHADOZ profiles, expressed as percent anomaly from annual mean, determined from the MLR model in the lower and middle FT (5-10 km: a, 10-15 km: b) and for the LMS (15-20 km: c). The tropopause Height (TH) seasonal cycle (d, in km) is based on the 380 K potential temperature surface from the radiosondes. Dots indicate monthly values; error bars display the 95% confidence intervals.

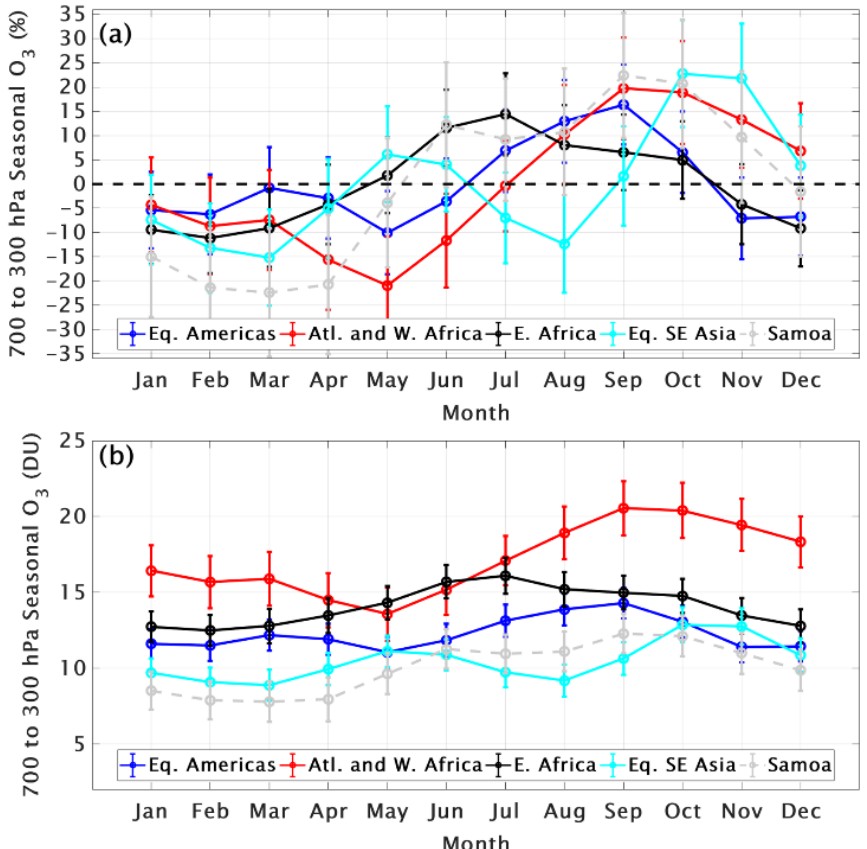

**Figure 6.** Monthly ozone variability for the five combined SHADOZ+IAGOS regions (defined in
**Figure 1** and **Table 3**), expressed as anomaly from annual mean in (a) percent with actual
values in DU (b), for FTp (700-300 hPa) column. Dots indicate the monthly values; error bars
display the 95% confidence intervals.


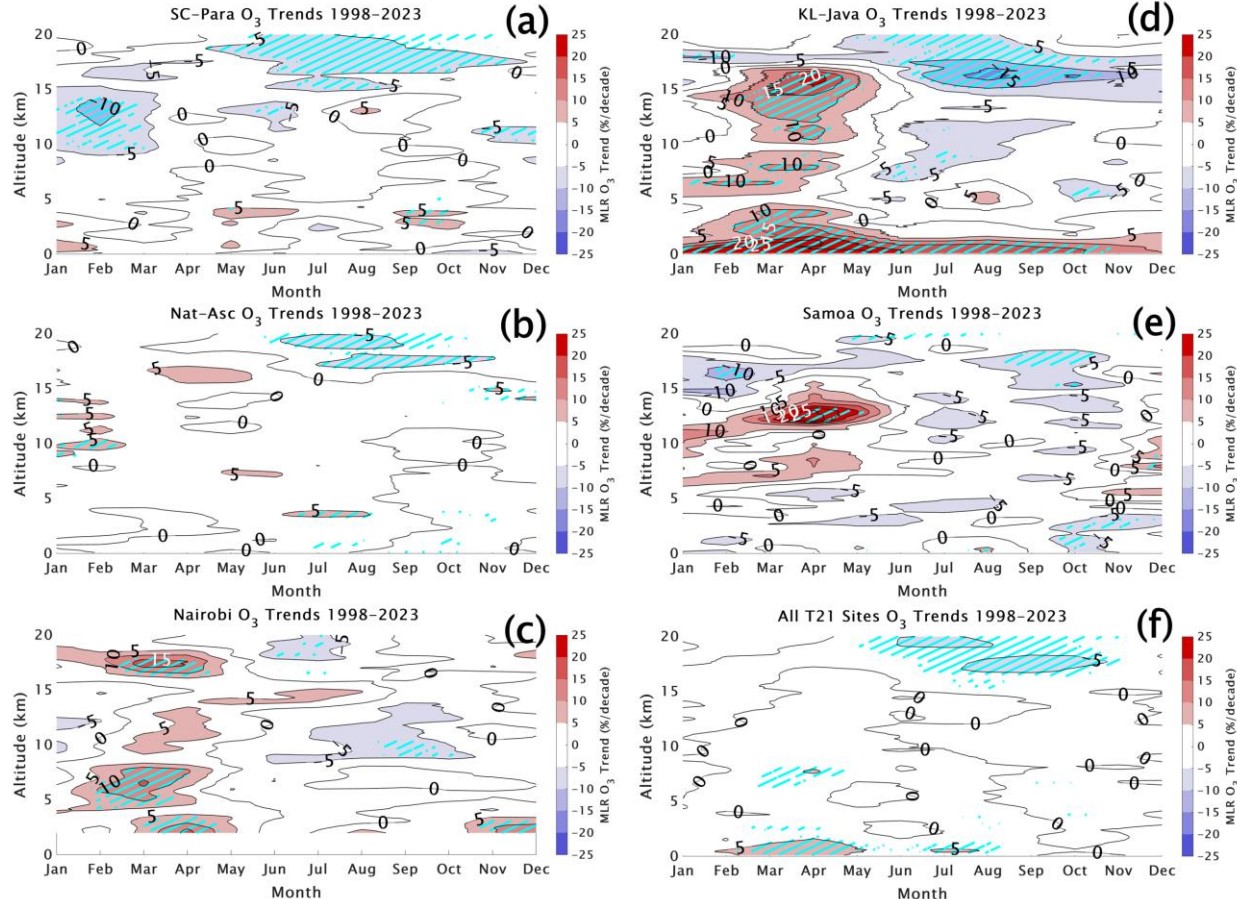

**Figure 7.** Monthly MLR ozone linear trends from 0 to 20 km in percent per decade for the SHADOZ T21
stations (a) San Cristóbal-Paramaribo (SC-Para); (b) Natal-Ascension (Nat-Asc) (c) Nairobi, (d) Kuala
Lumpur-Watukosek (KL-Java); (e) Samoa. This is an update to Figure 6 in T21. In (f), average trends
over (a) through (e) are displayed by combining the records from all eight individual T21 SHADOZ
stations. Positive trends are shown in red shades and negative trends are shown in blue shades. Trends
with p-values <0.05 (exceeding the 95% confidence interval) are shown with cyan hatching.

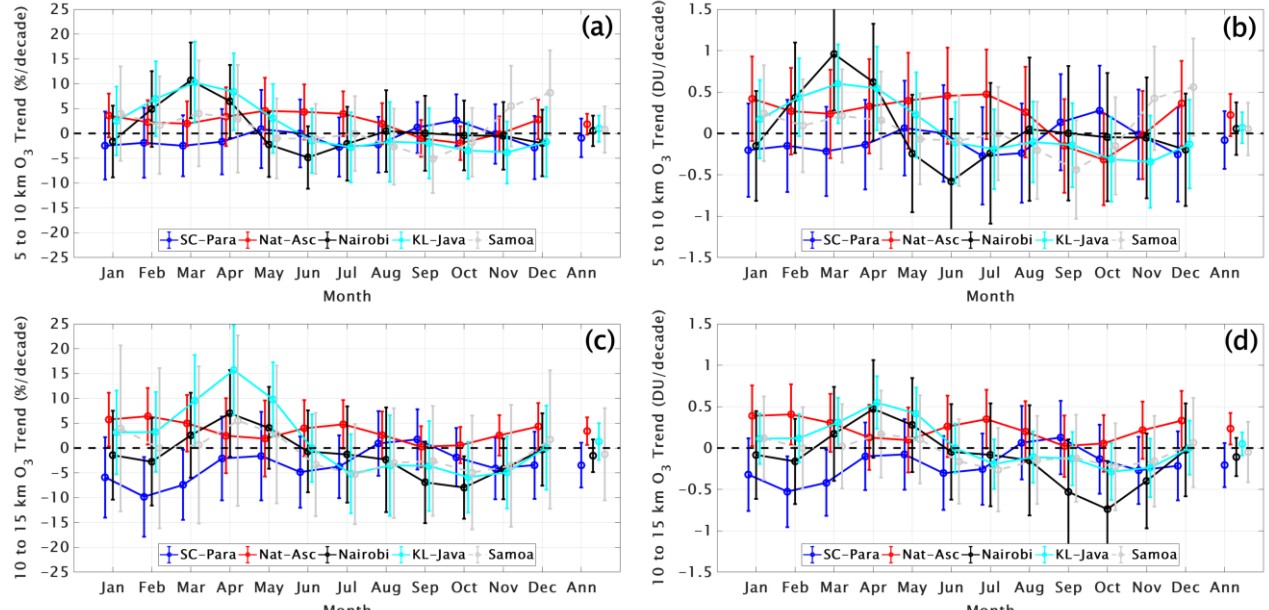

**Figure 8.** Monthly and annual MLR trends for five T21 SHADOZ sites in lower FT ozone column,
integrated from 5-10 km, for (a) %/decade (b) DU/decade; (c) and (d) same as (a) and (b)
respectively but for upper FT ozone column (10-15 km), derived from SHADOZ sondes. Dots
indicate the monthly and annual trends; error bars display the 95% confidence intervals.

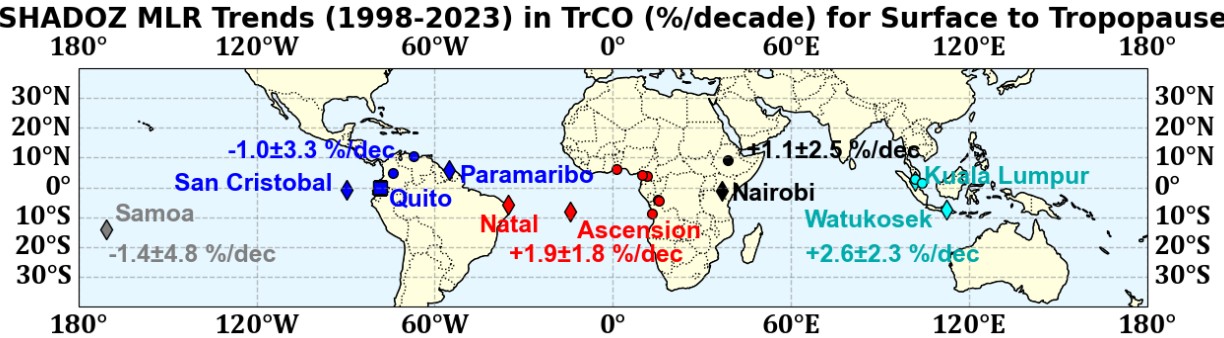

**Figure 9.** Summary map of the annual 1998-2023 MLR trends for five T21 SHADOZ sites for the total
tropospheric column ozone (TrCO), surface to tropopause, in %/decade from **Table 2** including the 95%
confidence intervals.

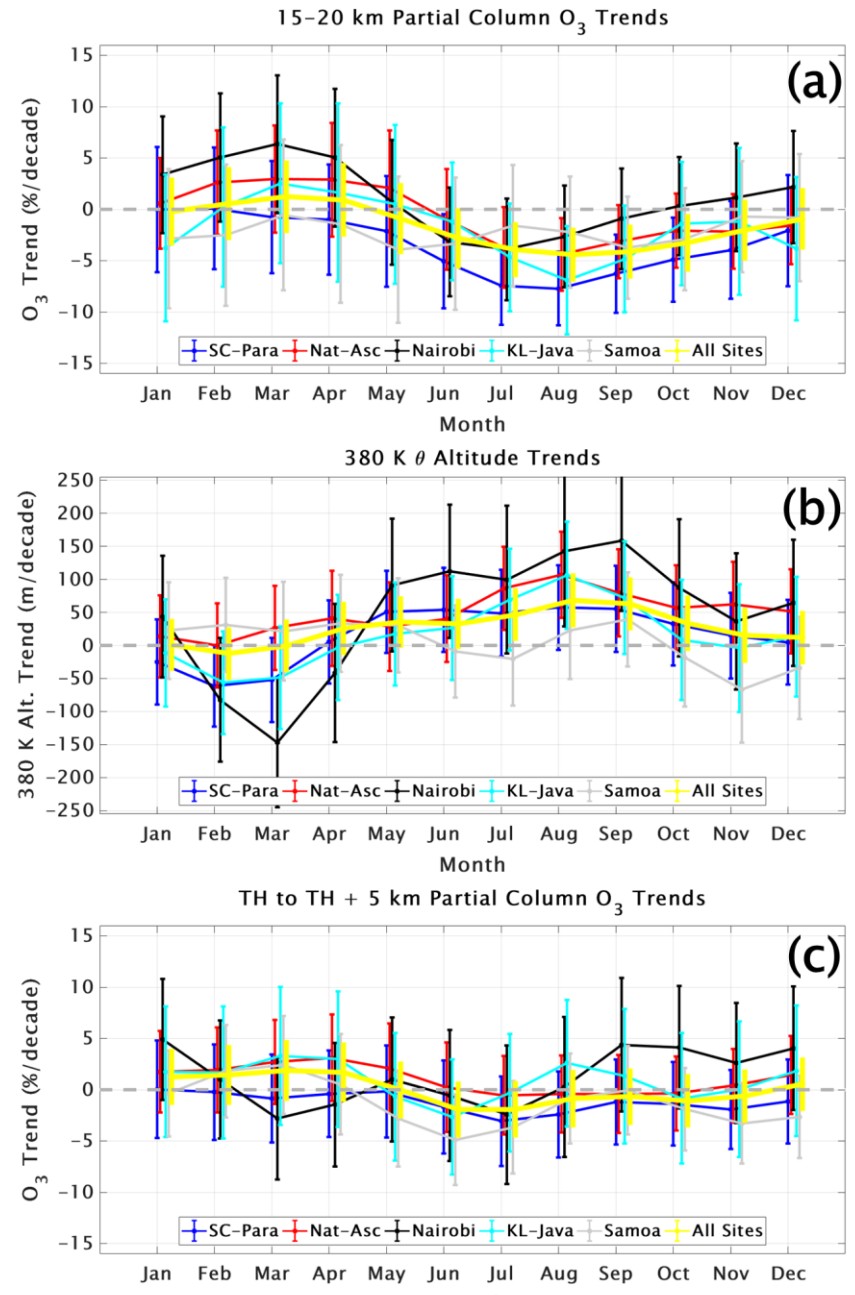

**Figure 10**. (a) Monthly MLR trends (colored dots) derived from SHADOZ T21 stations
highlighting a July-October decrease in LMS ozone in 15-20km layer; yellow dots denote the
mean of all T21 stations, with error bars indicating the 95% confidence intervals. (b)
Corresponding TH trends (380 K potential temperature; θ) derived from the radiosondes. (c)
Same as (a) except trends have been computed for the segments between the tropopause and 5
km above the TH. Compared to (a) the trends in the tropopause referenced ozone column (c)
become close to zero throughout the year.

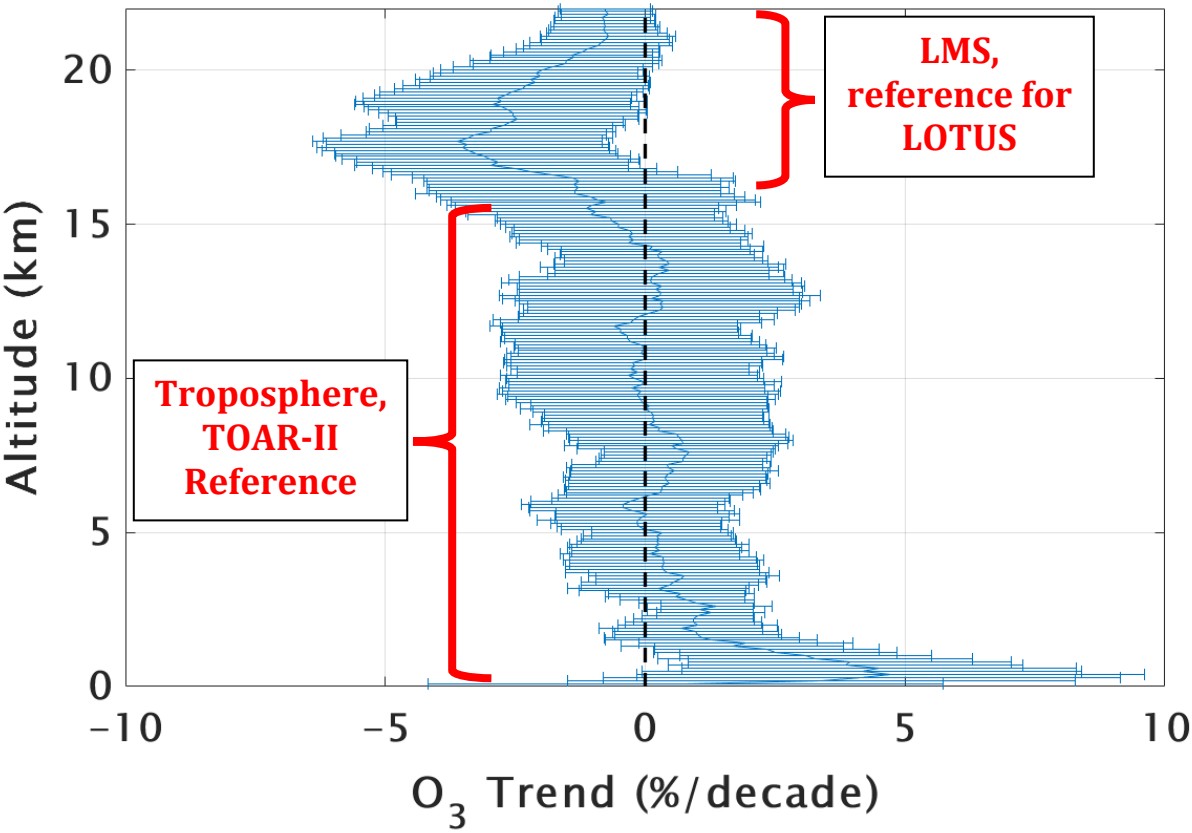

**Figure 11.** The total mean annual ozone trend (solid blue line), based on mixing ratio changes in 100-m intervals, from the surface to 22 km for all eight T21 SHADOZ profile datasets in %/decade with the 95% confidence interval range denoted. The LMS region of interest to the stratospheric community, e.g., the LOTUS activity, while the tropospheric segment is marked as the primary TOAR-II focus. The - 4%/decade trend in LMS ozone is similar to that derived from satellites in that region. The mean change throughout the FT is negligible and within the uncertainty range except below 2 km where mean increases ~+5%/decade are indicated. The near-surface trends are primarily a result of rapid increases in urbanized regions of equatorial SE Asia (Stauffer et al., 2024).

(a)

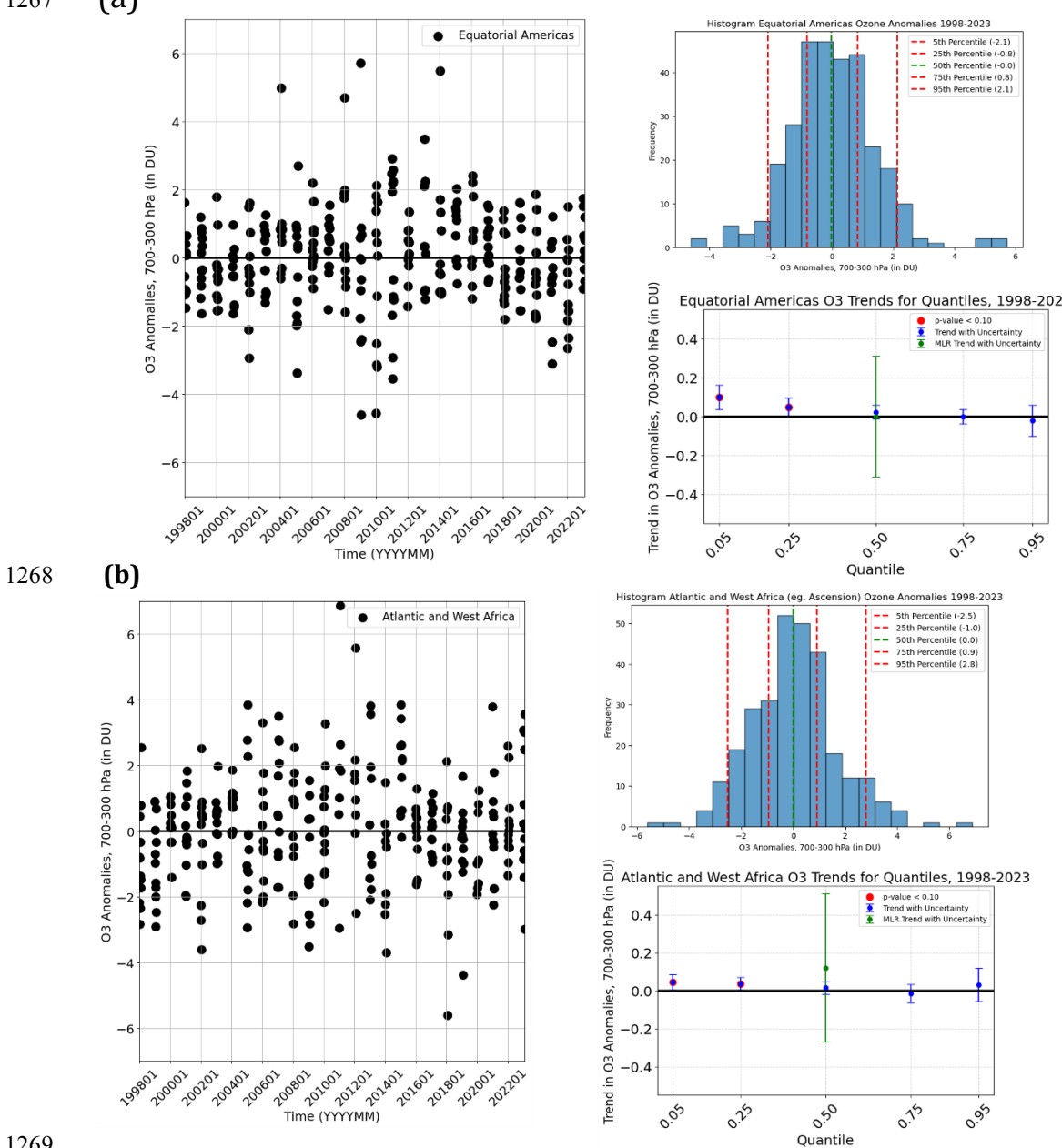

**(b)**

**(c)**

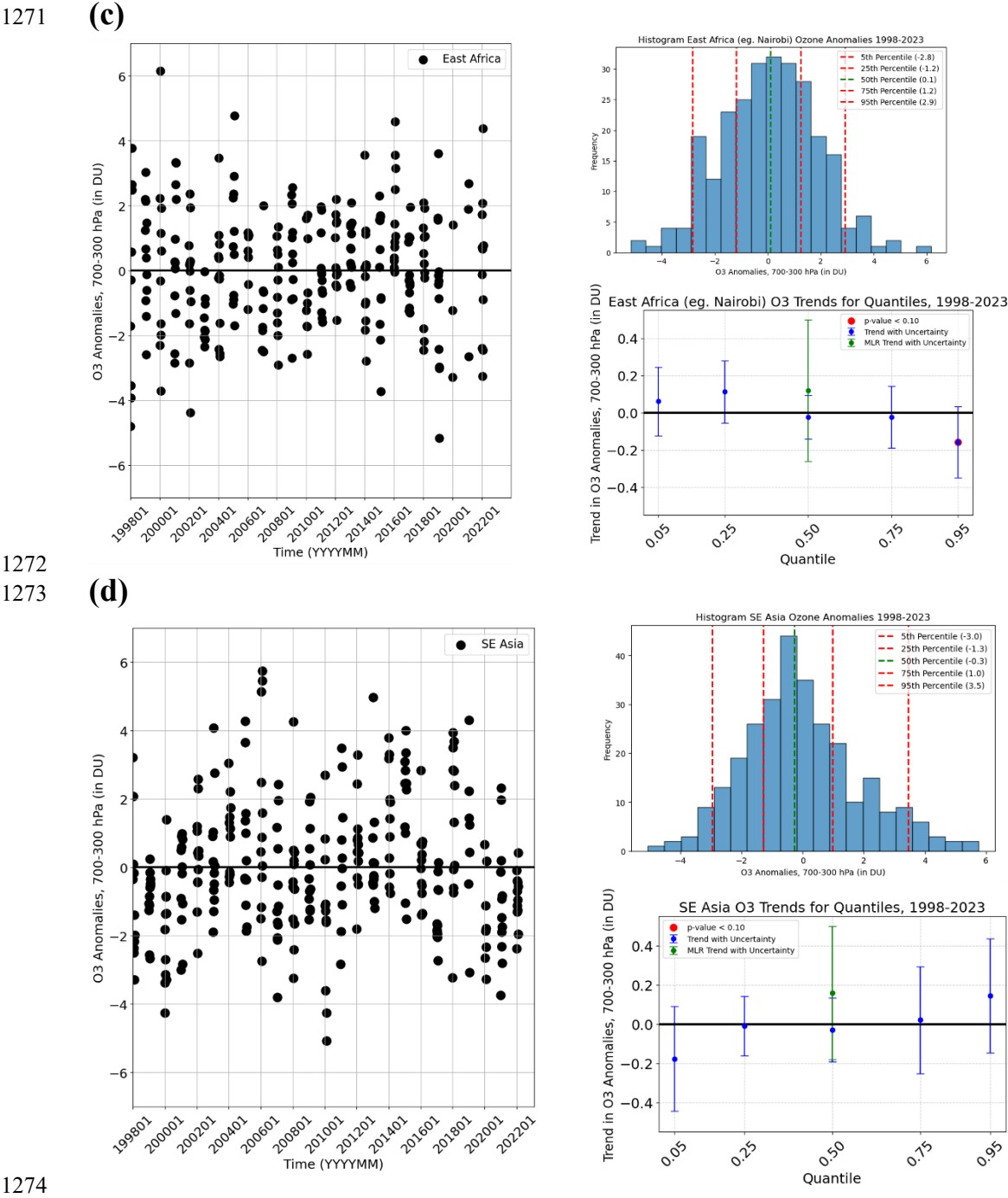

**(d)**
**Figure 12.** FTp column ozone segment anomalies (monthly means) from SHADOZ sondes and IAGOS data, as
described in **Table 3.** Each frame displays the anomalies timeseries in DU at the left, a histogram of those
anomalies in DU (each has a unique scale, ozone anomaly values for each percentile (dashed lines) are in
parentheses) and the trends distribution by quantiles. Trends from annual MLR ozone (median, 50%-ile) over
1998-2023 are green circles with their respective +/- 2sigma bars. The same period trends computed with QR by
quantiles (0.05= lowest 5%-ile), 0.25 (25%-ile), 0.50 for median trends, 0.75 (75%-ile) and 0.95 (95%-ile) are blue
circles.  Red circles denote p<0.10.  (a) equatorial Americas; (b) Atlantic and West Africa region; (c) east Africa; (d)
equatorial southeast Asia.

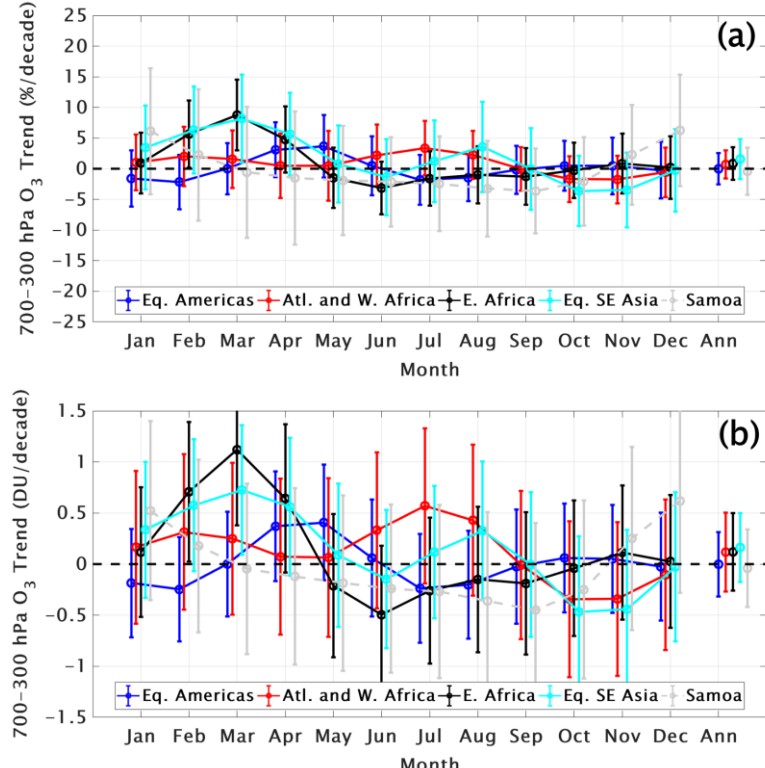

**Figure 13.** Monthly and annual MLR ozone trends for 5 combined SHADOZ+IAGOS regions, defined in **Table 3**, for FTp column in (a) %/decade and (b) DU/decade. Dots indicate the monthly and annual trends, whereas error bars display the 95% confidence intervals.

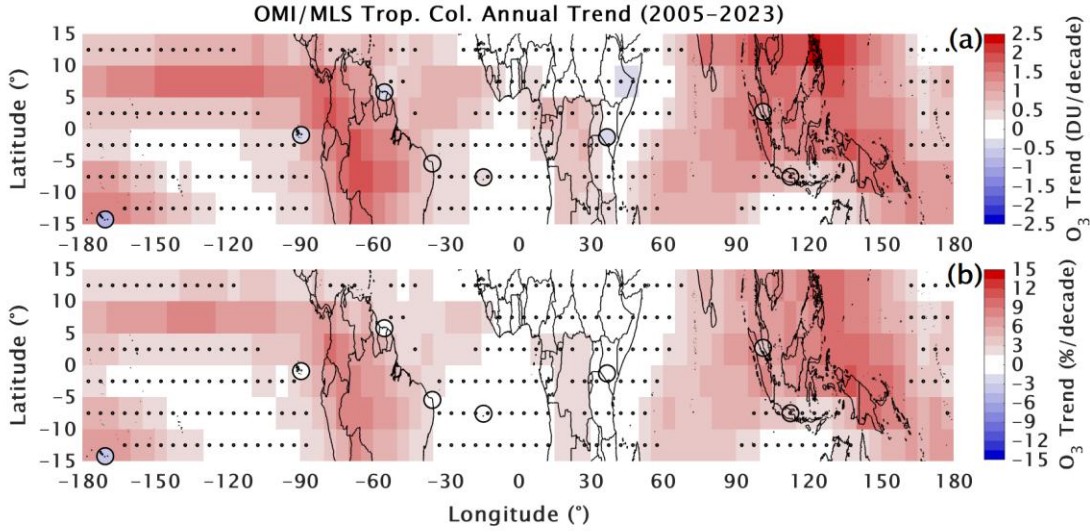

**Figure 14.** The most recent trends, 2005-2023, shown for the equatorial region based on updated OMI/MLS tropospheric total column ozone (TrCO$_{satellite}$) estimates in which a ~1% per decade positive drift in OMI was corrected. The corresponding SHADOZ-derived TrCO$_{sonde}$ column changes for 2005-2023 are superimposed on the map. Stippling indicates where OMI/MLS trends *do not* exceed the 95% confidence interval (i.e., historically referred to as statistically insignificant).

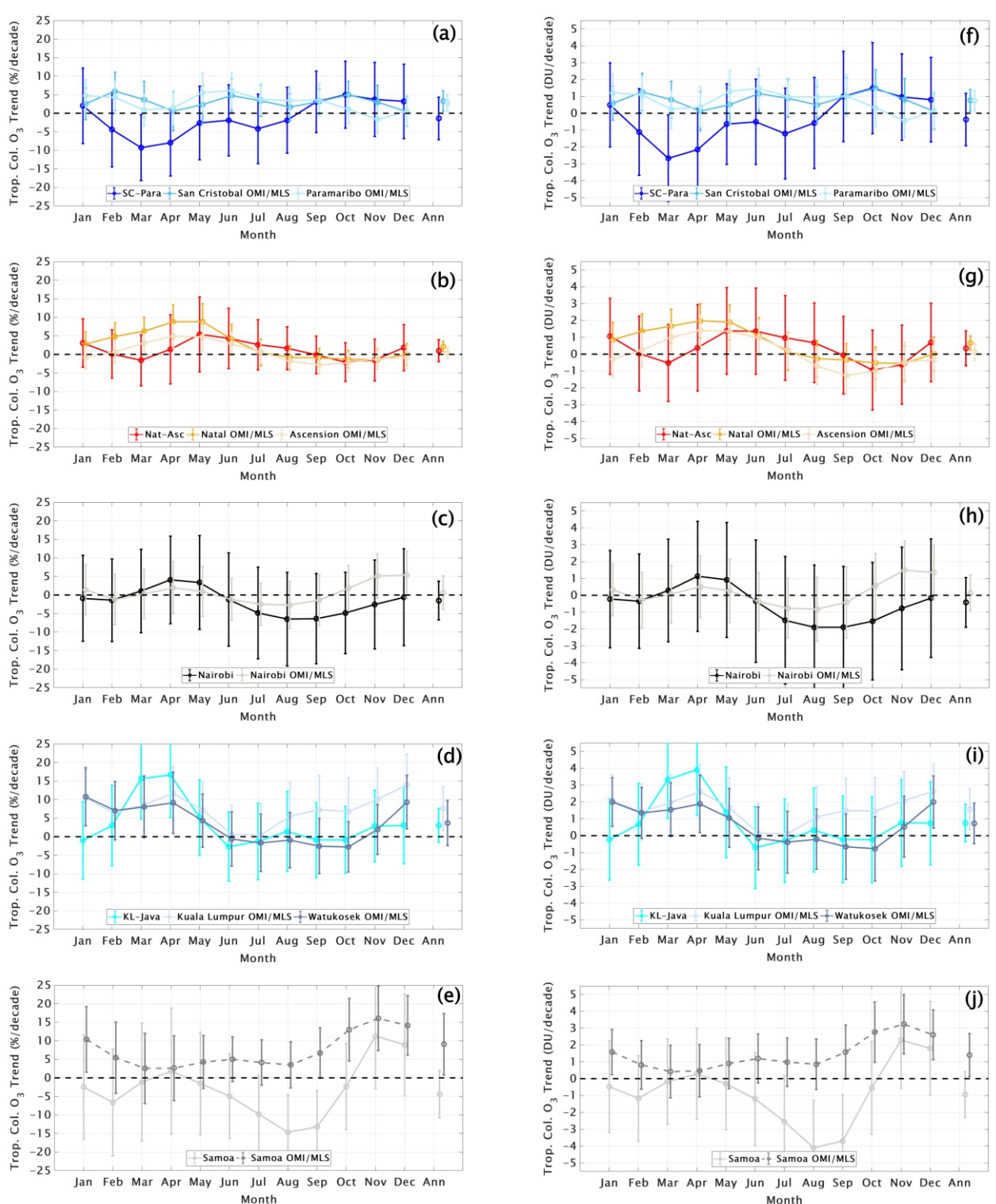

**Figure 15.** Monthly and annual MLR ozone trends in total tropospheric column (defined using the WMO lapse rate tropopause; TrCO) for the five T21 stations and the OMI/MLS pixel for each individual SHADOZ station each region. Dots indicate the ozone trend in % (a-e) and DU (f-j) per decade; error bars show the 95% confidence intervals.

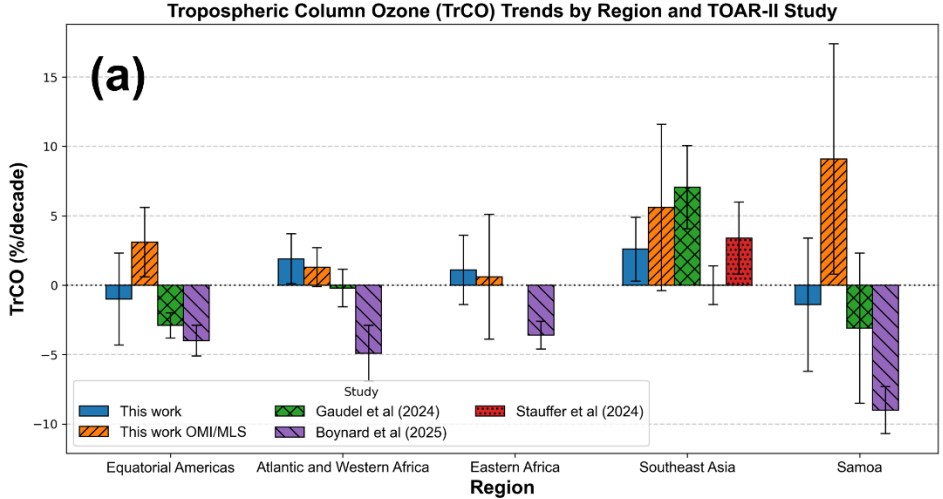

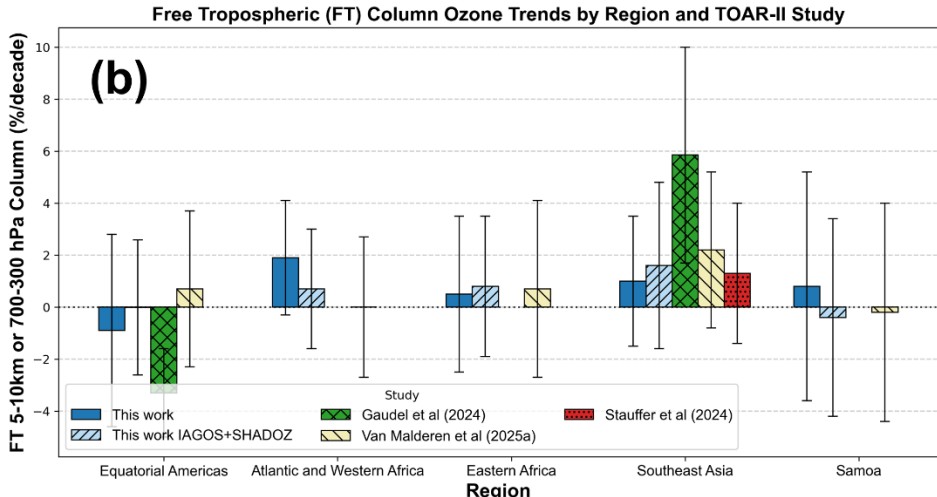

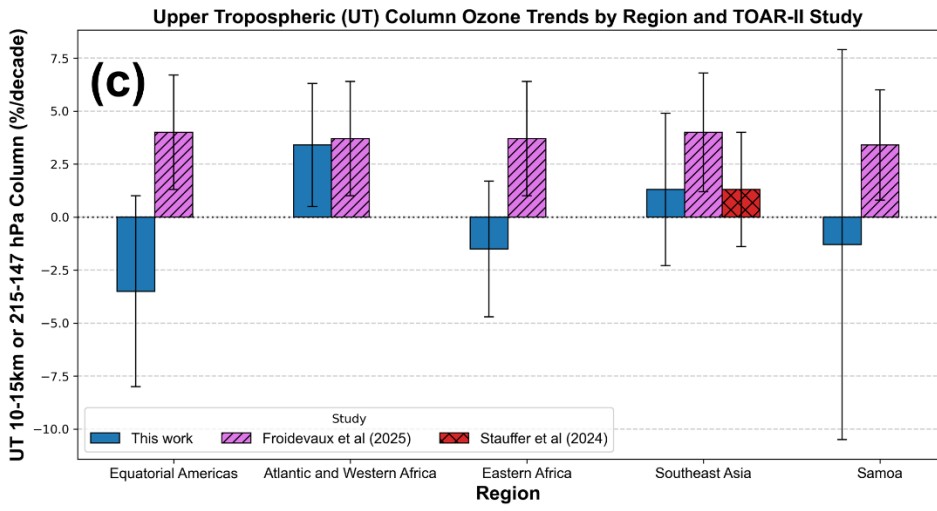

**Figure 16.** Summary of (a) TrCO, (b) FT, and (c) UT trends from multiple TOAR-II studies (detailed in Table 6). Trends are annual values in %/decade (sometimes estimated from each study); a different color represents each study across (a-c). Error bars show the 95% confidence intervals. If there are multiple values for a region in one study, a mean value is used.