# Peer review of "Tropical Tropospheric Ozone Trends (1998 to 2023): New Perspectives"

_EGUsphere, 2024_

## Referee Comment (RC2)

**Surface Tropical Ozone Trends (1998 to 2023): A Synthesis from SHADOZ, IAGOS and OMI/MLS Observations**

Anne M. Thompson, Ryan M. Stauffer, Debra E. Kollonige, Jerald R. Ziemke, María Cazorla, Pawel Wolff, Bastien Sauvage

**General comments**

The manuscript assesses the ozone trends in tropospheric columns and the lowermost stratosphere based on ozone profiles obtained from ozonesondes (SHADOZ) and aircraft measurements (IAGOS) within the tropics (15°N-15°S). The authors extended the period previously analyzed in Thompson et al., 2021 (1998-2019) until 2023. SHADOZ stations were merged into five zones to improve representativeness and assess trends using a multi-lineal regression model and quantile regression, the latter as suggested by TOAR-II guidelines. Ozone trends in the tropics are relevant in terms of radiative balance and ozone precursor emissions, to mention some. In addition, the article addresses topics of high interest, such as declining convection in some tropical regions and the trends in the lowermost stratosphere. However, in my assessment, this manuscript needs major revisions before considering for publication.

Beyond the potential value of updating the previous work until 2023, the scientific motivation for the research is unclear. Neither the abstract nor the introduction clearly establish the research's contribution. Many aspects discussed in the paper are not developed in depth in the introduction, and instead, the references are widespread throughout the document. This option is certainly valid, but it lacks focus. The questions outlined (four bullets) at the end of the introduction require further elaboration. Also, in the introduction more literature and key findings need to be discussed. The figures require improvement (e.g., Figures 1, 2, 3, 5 and 7), and some of them can be moved to supplemental material to reduce the current number and emphasize the key message. In my opinion, the current version requires better organization (e.g., Methodology and Summary), and some sections must be refined to describe better the results (e.g., the first paragraph of section 3.3.1).

**Minor comment**

Lines 30-34: In the first lines, I expect the presentation of the scientific gap the main motivation. Instead, the authors included three references, which can be added later in the introduction.

Line 52: Why not include a reference here?

Lines 65-66: Key references are missing here.

Line 84: "TOAR-II decided…" Is that a TOAR-II working group decision or a TOAR-II suggestion?

Line 88 Which phases of TOAR?

Lines 109-119: These four questions need more elaboration and a clear statement of the scientific objectives. In what way could this comparison be different from other studies published in the frame on TOAR-II? Throughout the manuscript, the authors addressed other

publications such as Gaudel (2024) or Van Malderen (2024). However, at this point, we should be clear about the objectives.

Line 132: P-T-U is not defined for readers unfamiliar with this term.

133-134: The authors are referencing Quito station using three papers; the lat lon or something else changes within the publication of these papers?

Lines 135-137: What has changed in these three references?

Line 170: Note that LMS was already defined in line 145

Line 207 Complete: SOTC (ref)

Line 175: Define (P-T-U) earlier

262: Add ozone: tropospheric ozone trends.

277-279: The wording of this sentence is not clear to me: *"In the individual site analyses of HEGIFTOM observations (Van Malderen et al., 2024a), where all individual ozone records (L1) and monthly means (L3) were analyzed, annually averaged trends usually turned out to be the same within uncertainties".* Can L1 and L3 be defined explicitly in the main text?

Line 289: The mention of the Walker circulation is a good example of the kind of background needed to improve the introduction.

Line 299: This sentence is already mentioned in the methodology: *We assign TH to the altitude of the 380K potential temperature.*

Line 299: If trends of the LMS are one of the relevant topics addressed in this research, why is Randel (2007) not included in the background discussion in the introduction?

Line 314: I read cf. here and in other parts of the article. What is the meaning here?

Line 339-341: I understand that this interpretation is valid for 5-10 km. Is it similar to other portions of the troposphere (e.g., upper troposphere)?

Line 437: Instead of "sufficient" I suggest representative.

Line 463: I suggest defining OLR to avoid ambiguities

Lines 476-480: If these lines represent the authors' conclusions, shouldn't they also be paraphrased in the abstract? The word "definitive" isn't too strong?

---

## Author Comment (AC1)

**Authors' Response to RC1**: 'Comment on egusphere-2024-3761', Anonymous Referee #1, 15 Apr 2025

Very good paper. It should be accepted with minor revisions as presented below: *Thank you for the compliment. In Response to RC 2 we modified the paper for more focus, clarity of messages and, we hope, more insight and impact in understanding tropospheric ozone trends near the Equator over the past ~25 years. We hope these clear up some of your questions (e.g., around Ll 290). The manuscript changes were also instigated by a number of TOAR II articles published or posted as preprints since our original submittal and that also analyze ozone observations from ground-based (GB) and satellite data. The two principal modifications in our revised paper are (1) a more thorough background in the Introduction, illustrating the necessity and motivation for a study that turns out to solidify the advantages of rigorously reprocessed GB data for trends analysis over shorter, less certain satellite products that diverge from GB measurements and from one another; (2) more details on the sensitivity studies we conducted (a) varying profile Sample Numbers in the free troposphere (FT); (b) evaluating of QR and MLR statistical approaches to trends; (c) varying start and end times of the GB time-series. A significant new result in the latter calculations is that trends determined from 12 years in either GB or satellite trends are so uncertain compared to 20-25-year trends that they are not reliable for the needs of TOAR II.*

*Our Summary is longer and better links our study to other TOAR II papers. We modified the title and added several references.*

Line 55: missing reference *Reference has been added. Thank you.*

Line 200/201: How is the tropopause defined? It would be interesting to address how the tropopause is found in the paper. *We specify that the WMO lapse rate convention is followed. This can be found at WMO, 1957: Definition of the tropopause. WMO Bull., **6**, 136*

Line 207: Gaudel et al., 2024 the reference does not open with the link provided. Please provide the correct link. https://acp.copernicus.org/articles/24/9975/2024/

Line 292-295: Explain further how you arrived at this analysis. *See above. More details on the climatology appear in T21 (see their Figure 2)*

**References:** I suggest reviewing the references according to the format proposed by the journal. *All references entered as per journal format*

---

## Author Comment (AC2)

**Surface Tropical Ozone Trends (1998 to 2023): A Synthesis from SHADOZ, IAGOS and OMI/MLS Observations**

Anne M. Thompson, Ryan M. Stauffer, Debra E. Kollonige, Jerald R. Ziemke, María Cazorla, Pawel Wolff, Bastien Sauvage                                    15 July 25

**General comments**

The manuscript assesses the ozone trends in tropospheric columns and the lowermost stratosphere based on ozone profiles obtained from ozonesondes (SHADOZ) and aircraft measurements (IAGOS) within the tropics (15°N-15°S). The authors extended the period previously analyzed in Thompson et al., 2021 (1998-2019) until 2023. SHADOZ stations were merged into five zones to improve representativeness and assess trends using a multilineal regression model and quantile regression, the latter as suggested by TOAR-II guidelines. Ozone trends in the tropics are relevant in terms of radiative balance and ozone precursor emissions, to mention some. In addition, the article addresses topics of high interest, such as declining convection in some tropical regions and the trends in the lowermost stratosphere. However, in my assessment, this manuscript needs major revisions before considering for publication.

Beyond the potential value of updating the previous work until 2023, the scientific motivation for the research is unclear. Neither the abstract nor the introduction clearly establish the research's contribution. Many aspects discussed in the paper are not developed in depth in the introduction, and instead, the references are widespread throughout the document. This option is certainly valid, but it lacks focus. The questions outlined (four bullets) at the end of the introduction require further elaboration. Also, in the introduction more literature and key findings need to be discussed. The figures require improvement (e.g., Figures 1, 2, 3, 5 and 7), and some of them can be moved to supplemental material to reduce the current number and emphasize the key message. In my opinion, the current version requires better organization (e.g., Methodology and Summary), and some sections must be refined to describe better the results (e.g., the first paragraph of section 3.3.1).

**General Responses**

*Please note that title above is not correct.*

**Paper Focus – Framing the Questions & Research Outline.** *Thank you to the Reviewer. You are correct about uncertain paper focus. There were practical reasons for that.*

*First, August to November (2024) we were simultaneously writing this paper and completing half or more of the analyses and graphics for the TOAR/**HEGIFTOM paper #1*** *(Van Malderen, Thompson, Kollonige, Stauffer et al.,* https://doi.org/10.5194/acp-25-7187-2025*). We also were active in formulating HEGIFTOM -2, i.e.,Van Malderen et al, in press: egusphere-2024-3745***.** *In those two papers the same SHADOZ profile data are used but "troposphere"-defined columns differed. For example, to have a uniform pressure-defined tropopause globally, "HEGIFTOM-1" capped the tropopause at all 55 sites analyzed pole-to pole at 300 hPa, approximately 1/3 vertical extent below the tropical tropopause of SHADOZ profiles. Statistical protocols and interpretation of the TOAR guidelines on trend methods and required output varied somewhat between the two papers. HEGIFTOM-1 omitted 3 of 14 long-term SHADOZ stations. The analyses (i.e., figures and Table) for HEGIFTOM-1 (some were iterated by "consensus" with 10-15 prime authors) changed several times over the August to November 2024 period. Thus, which material would appear in which paper became a moving target, hard to resolve as the submittal deadline for TOAR II contributions came closer. Note that TOAR II required that contributions for consideration in the TOAR II official "Assessment papers" be*

*published in Copernicus journals and submitted NLT 30 November 2024 so there was not time for the lead authors to iterate. It turns out that the trend results in the present paper are nearly the same as HEGIFTOM-1 although the periods differ slightly, 1998-2023 here, vs 2000-2022 in HEGIFTOM-1. (This can be seen in our new Table 6). Due to the tight deadline, for This Study, we did not have time to contact all SHADOZ data providers. That has been corrected in this Submittal with a full set of authors.*

*Second, as more TOAR papers appeared on the Copernicus website several issues arose that greatly increased motivation for the current paper and caused us to undertake analyses beyond the original scope of the paper.*

*What was done to clarify the TOAR II context and to sharpen focus?  Three things:*

*(1) Amplify background and motivation for our paper. This consists of comparing our T21 work with Gaudel et al. (2024) and HEGIFTOM-1 (2025). However, several other TOAR II papers have been published or are in open review that highlight inadequacies in tropospheric ozone satellite estimates of trends, e.g., Pope et al., 2023; Froidevaux et al., 2025; Keppens et al., 2025. We also discuss the latter because it is essential to point out errors or limitations in those studies.*

*(2) Divide Results and Discussion into 2 parts, the first a T21 update that includes comparing trends for individual SHADOZ stations with those from OMI/MLS for 2005-2023.*

*(3) In the second part (Section 2.2.3 in Methods and 3.5 in Results and Discussion) three statistical issues raised in TOAR II are examined in: (a) comparison of the two trends models used in most TOAR II studies: Quantile Regression and Multiple-Linear Regression; (b) sample numbers – does adding more (or removing) data increase (decrease) magnitude of FT ozone trends and/or their uncertainties? Here SHADOZ profiles were augmented with nearby IAGOS profiles. (c) how sensitive are the SHADOZ trends for individual stations (as in HEGIFTOM-1) to length of time-series, as short as 12 years for some satellite time-series, vs 20-30 years for ground-based data, e.g., 26 years for 12 SHADOZ stations?*

*The result is re-organization of the paper in: Section 1, Introduction (much more Background) and two sets of questions and goals) with additional references; Section 3, Results and Discussion, that is organized to present T21 updates followed by findings on the statistical TOAR II questions. The updated Summary and Conclusions (Section 4) compares our Results with those of three closely related TOAR II studies, articulating a consensus set of messages for TOAR II as a whole.* **IN SUMMARY, the revised paper integrates perspectives from SHADOZ, IAGOS and the longest-term satellite (OMI/MLS) record to address TOAR concerns about trends and uncertainties. We provide analyses of tropical ozone throughout FT and LMS, providing the "Reference" sonde trends from the highest-quality data \*and\* demonstrating that SHADOZ supplies sufficient sample numbers. The paper has a new Title.**

*\* Van Malderen, R., Thompson, A. M., Kollonige, D. E., Stauffer, R. M., Smit, H. G. J., Maillard Barras, E., Vigouroux, C., Petropavlovskikh, I., Leblanc, T., Thouret, V., Wolff, P., Effertz, P., Tarasick, D. W., Poyraz, D., Ancellet, G., De Backer, M.-R., Evan, S., Flood, V., Frey, M. M., Hannigan, J. W., Hernandez, J. L., Iarlori, M., Johnson, B. J., Jones, N., Kivi, R., Mahieu, E., McConville, G., Müller, K., Nagahama, T., Notholt, J., Piters, A., Prats, N., Querel, R., Smale, D., Steinbrecht, W., Strong, K., and Sussmann, R.: Global ground-based tropospheric ozone measurements: reference data and individual site trends (2000–2022) from the TOAR-II/HEGIFTOM project, Atmos. Chem. Phys., 25, 7187–7225, https://doi.org/10.5194/acp-25-7187-2025, 2025*

*\*\* Van Malderen, R., Zang, Z., Chang, K.-L., Björklund, R., Cooper, O. R., Liu, J., Barras, E. M., Vigouroux, C., Petropavlovskikh, I., Leblanc, T., Thouret, V., Wolff, P., Effertz, P., Gaudel, A., Tarasick, D. W., Smit, H. G. J., Thompson, A. M., Stauffer, R. M., Kollonige, D. E., Poyraz, D., Ancellet, G., De Backer, M.-R., Frey, M. M.,*

*Hannigan, J. W., Hernandez, J. L., Johnson, B. J., Jones, N., Kivi, R., Mahieu, E., Morino, I., McConville, G., Müller, G., Murata, I., Notholt, J., Piters, A., Prignon, M., Querel, R., Rizi, V., Smale, D., Steinbrecht, W., Strong, K., and Sussmann, R.: Ground-based tropospheric ozone measurements: Regional tropospheric ozone column trends from the TOAR-II/ HEGIFTOM homogenized datasets, DOI: 10.5194/egusphere-2024-3745, in press, 2025*

**Minor comment**

Lines 30-34: In the first lines, I expect the presentation of the scientific gap the main motivation. Instead, the authors included three references, which can be added later in the introduction. *Motivation is now clarified; namely, the importance of quantifying tropospheric ozone trends is spelled as well as the inadequacy of current satellite products for doing so.*

Line 52: Why not include a reference here? *Good point. Several from the 80s and 90s were relevant. We picked two: Schwartzkopf and Ramaswamy (1993) & Lacis et al. (1990)*

Lines 65-66: Key references are missing here *This section has been amplified and a number of new studies referenced. Further context is given in a separate section (now 1.1) on the TOAR project and the challenges in having suitable data for tropospheric ozone assessment in both TOAR I and the newer TOAR II. The limitations of satellite observations that were prominent in Gaudel et al. (2018, TOAR I) persisted in comparisons with ground-based data in Gaudel et al. (2024). Additional examples of limitations of satellite data are seen in Froidevaux et al. (2025), Boynard et al. (2025) and Keppens et al. (2025). These are spelled out as further motivation for why TOAR II must give serious consideration to trends based on ground-based data. See also the last Response below on use of the word "definitive."*

*Boynard et al: Tropospheric Ozone Assessment Report (TOAR): 16-year ozone trends from the IASI climate data record, https://doi.org/10.5194/egusphere-2025-1054, in review, 2025.*

*Froidevaux et al: Tropical upper-tropospheric trends in ozone and carbon monoxide (2005–2020): observational and model results, Atmos. Chem., Phys., https://doi.org/10.5194/acp-25-597-2025, 2025*

*Keppens et al: Harmonisation of sixteen tropospheric ozone satellite data records, egusphere-2024-3746, in review, 2025*

Line 84: "TOAR-II decided…" Is that a TOAR-II working group decision or a TOAR-II suggestion? *This is rephrased in Section 1.2, lines 158-173.*

Line 88 Which phases of TOAR? *Same as prior response, for Line 84*

Lines 109-119: These four questions need more elaboration and a clear statement of the scientific objectives. In what way could this comparison be different from other studies published in the frame on TOAR-II? Throughout the manuscript, the authors addressed other publications such as Gaudel (2024) or Van Malderen (2024). However, at this point, we should be clear about the objectives. *The reframing of the paper motivation with two broad goals – first, a T21 update and second, additional analyses for TOAR II statistical questions and comparisons with other papers – is now explicit and should give the paper more rigor and impact. There are three "update" and three "TOAR II" type questions in the revision.*

Line 132: P-T-U is not defined for readers unfamiliar with this term. *Thank you- it is now spelled out*

133-134: The authors are referencing Quito station using three papers; the lat lon or something else changes within the publication of these papers? *Because this paper introduces Quito as a SHADOZ station, we retained three references.*

Lines 135-137: What has changed in these three references? *Same as response for Line 133-134.*

Line 170: Note that LMS was already defined in line 145 *Thank you for pointing that out. We removed the redundant definition*

Line 207 Complete: SOTC (ref) *State of the Climate Report – Now removed as not needed*

Line 175: Define (P-T-U) earlier *Done, as above*

262: Add ozone: tropospheric ozone trends. *Corrected – thank you*

277-279: The wording of this sentence is not clear to me: *"In the individual site analyses of HEGIFTOM observations (Van Malderen et al., 2024a), where all individual ozone records (L1) and monthly means (L3) were analyzed, annually averaged trends usually turned out to be the same within uncertainties"*. Can L1 and L3 be defined explicitly in the main text? *Modified to "In the individual site analyses of HEGIFTOM observations (HEGIFTOM-1), where all individual ozone records (designated as L1) and monthly means (denoted as L3) were analyzed, annually averaged trends turned out to be the same within uncertainties. Where QR is applied in the present study, L1 ozone data are used"*

Line 289: The mention of the Walker circulation is a good example of the kind of background needed to improve the introduction. *Now appears around Line 130*

Line 299: This sentence is already mentioned in the methodology: *We assign TH to the altitude of the 380K potential temperature. Thank you. Redundancy removed*

Line 299: If trends of the LMS are one of the relevant topics addressed in this research, why is Randel (2007) not included in the background discussion in the introduction? *Is now added*

Line 314: I read cf. here and in other parts of the article. What is the meaning here? *Academically used from the Latin meaning "compared". All occurrences deleted*

Line 339-341: I understand that this interpretation is valid for 5-10 km. Is it similar to other portions of the troposphere (e.g., upper troposphere)? Line 437: Instead of "sufficient" I suggest representative. *"Sufficient" has been retained because the TOAR II Committee comments on a number of papers in review, as well as papers like HEGIFTOM-1 and Gaudel et al. (2024), repeatedly claim that 2 profiles/month are inadequate for free or full tropospheric trends. TOAR II workshop discussions (the most recent on 23 June 2025) continue to stress large sample numbers, preferably 8-15/month for the FT profiles. HEGIFTOM-1 reduced their trend analysis datasets to 2/month (see Responses to Reviewers for egusphere-2024-3736) with no significant change in trends and This Study increased the samples numbers by factors of 1.3-2.5 for 4 tropical regions – still no change in median trends. (In both cases, increased uncertainties). Our two studies show that the sampling frequency arguments for tropospheric ozone are over-stated for the global ground-based data, not only for the tropical.* **The "sufficiency"of the SHADOZ and other sonde records is a very important message to the TOAR-II community and beyond.**

Line 463: I suggest defining OLR to avoid ambiguities *Done-spelled out*

Lines 476-480: If these lines represent the authors' conclusions, shouldn't they also be paraphrased in the abstract? *Correct. Thank you for the reminder. Additional analyses have provided important new results. The T21 updated results are summarized in the revised Abstract and the newer statistically oriented results follow.* The word "definitive" isn't too strong? *Correct. Abstract is substantially re-written with conclusions numbered for emphasis. We assert that "definitive" is not too strong. One could argue that the ozone trends of HEGIFTOM-1 (that uses the word "Reference" in the title) deserve that adjective. However, the HEGIFTOM-1 trends (1) exclude 2 SHADOZ stations with high-quality data; (2) their trends ending at 300 hPa omit much of the tropical troposphere; (3) tabulate only QR and MLR annual means, displaying few seasonal trends. The present paper displays and makes available monthly mean trends (in 100-m segments) for the five equatorial sites determined from MLR \*as well as\* the corresponding MLR and QR annual means. Furthermore, a new **Fig. 11** illustrates several cases where MLR and QR medians diverge slightly (though with overlapping uncertainties) and how the corresponding quantile trends from QR provide information complementary to MLR. **Figs. 7-9** from MLR give unique insight into seasonal changes whereas QR distinguishes trends among lower- vs higher ozone populations/distributions. However, the latter points are 'details'. What IS important is that the two HEGIFTOM papers plus This Study provide higher quality data as a whole than the current suite of satellite products or models. Collectively the three papers (and Gaudel et al. [2024] in most cases) are* **"definitive" in providing the best data available for a tropospheric ozone assessment. HEGIFTOM-1, HEGIFTOM-2 and This Study constitute measuring sticks whereby satellite products and models must be evaluated. The integrity of these ground-based trends must be recognized and not discounted in the overall TOAR II Assessment.**

*A new* **Table 6** *compares ozone trends and uncertainties for 4 tropical regions and Samoa as computed from four ground-based data studies (Gaudel et al., 2024; HEGIFTOM-1 [2025]; HEGIFTOM-2 [2025]; This Study) for: Free Troposphere (FT); Upper Troposphere (UT); Total Column (Tr) troposphere. The time spans range between 1995 and 2023. In general, the results for these periods are consistent across the four studies. Trends for the 3 column segments range from ~(-3 to+3)%/decade. The greatest changes only occur over southeast Asia, driven by boundary layer increases (~5-6%/decade). The OMI/MLS trends (2005-2023) generally support this picture (except at Samoa); the Froidevaux et al. (2025) UT increases, based on MLS observations that disagree with the corresponding sonde measurements, are too high.*